# Genetic inactivation of mTORC1 or mTORC2 in neurons reveals distinct functions in glutamatergic synaptic transmission

Matthew P McCabe[†], Erin R Cullen[†], Caitlynn M Barrows, Amy N Shore, Katherine I Tooke, Kathryn A Laprade, James M Stafford, Matthew C Weston*

University of Vermont, Department of Neurological Sciences, Burlington, United States

**Abstract** Although mTOR signaling is known as a broad regulator of cell growth and proliferation, in neurons it regulates synaptic transmission, which is thought to be a major mechanism through which altered mTOR signaling leads to neurological disease. Although previous studies have delineated postsynaptic roles for mTOR, whether it regulates presynaptic function is largely unknown. Moreover, the mTOR kinase operates in two complexes, mTORC1 and mTORC2, suggesting that mTOR's role in synaptic transmission may be complex-specific. To better understand their roles in synaptic transmission, we genetically inactivated mTORC1 or mTORC2 in cultured mouse glutamatergic hippocampal neurons. Inactivation of either complex reduced neuron growth and evoked EPSCs (eEPSCs), however, the effects of mTORC1 on eEPSCs were postsynaptic and the effects of mTORC2 were presynaptic. Despite postsynaptic inhibition of evoked release, mTORC1 inactivation enhanced spontaneous vesicle fusion and replenishment, suggesting that mTORC1 and mTORC2 differentially modulate postsynaptic responsiveness and presynaptic release to optimize glutamatergic synaptic transmission.

*For correspondence:
mcweston@uvm.edu

[†]These authors contributed equally to this work

Competing interests: The authors declare that no competing interests exist.

## Introduction

The mechanistic target of rapamycin (mTOR) signaling network is an evolutionarily conserved group of interacting proteins centered around the ubiquitously expressed serine/threonine kinase mTOR. In a variety of species and cell types, mTOR signaling regulates fundamental cell biological processes such as cell growth, survival, and division (*Saxton and Sabatini, 2017*). In the nervous system, however, mTOR signaling plays a more specific role in neuronal communication by tuning the strength of synaptic connections (*Henry et al., 2012*; *Huang et al., 2013*; *Niere et al., 2016*; *Sperow et al., 2012*; *Weston et al., 2014*). This regulation of synaptic strength by mTOR is thought to be necessary for learning and memory, setting E-I balance, and maintaining synaptic homeostasis. Furthermore, variants in several genes in the mTOR signaling network cause neurological diseases including epilepsy and autism (*Crino, 2011*; *Lipton and Sahin, 2014*), and increasing evidence suggests that dysregulation of synaptic transmission is a key feature of these diseases (*Zoghbi and Bear, 2012*).

A broad distinction in the organization of the mTOR signaling network is that the mTOR kinase operates in two multiprotein complexes, mTORC1 and mTORC2 (*Hay and Sonenberg, 2004*). Both the substrates of mTOR and the downstream cellular processes it affects are different depending on its association with mTORC1 or mTORC2 (*Wullschleger et al., 2006*). Biochemically, mTORC1 and mTORC2 are distinguished by their protein composition. Although they both contain mTOR and other components such as Deptor and mLST8, Raptor is a protein that is unique to mTORC1 and is essential to its function, whereas Rictor is unique to mTORC2 and essential to its function.

Pharmacologically, the mTOR inhibitor rapamycin and its derivatives are more effective at inhibiting mTORC1 than mTORC2 (*Kang et al., 2013*), but this distinction is lessened with longer exposure and at higher concentrations (*Sarbassov et al., 2006*).

Previous studies have demonstrated changes in postsynaptic function by manipulating mTOR signaling either pharmacologically or genetically. Because of the link between mTOR, protein synthesis, and long-term synaptic plasticity, several studies have shown the necessity for intact mTORC1 and mTORC2 function in long-term potentiation and long-term depression (*Huang et al., 2013*; *Stoica et al., 2011*; *Tang et al., 2002*). At the molecular level, mTOR signaling and its downstream targets are known to regulate AMPA receptor expression and synapse number (*Ran et al., 2013*; *Wang et al., 2006*). More recently, postsynaptic loss of mTORC1 was shown to reduce evoked excitatory postsynaptic current (eEPSC) amplitudes onto Purkinje neurons, but mTORC2 loss did not, suggesting specific roles for the two complexes in the regulation of synaptic transmission (*Angliker et al., 2015*).

Despite progress in delineating the regulation of postsynaptic function by mTOR, the role of mTOR in presynaptic function, and more specifically synaptic vesicle (SV) release, is largely unexplored. Studies at the *Drosophila* neuromuscular junction and in rat hippocampal neurons have shown that postsynaptic mTORC1 activity provides a retrograde signal that enhances the readily releasable pool (RRP) of SVs in response to a reduction in postsynaptic glutamate receptor activity (*Henry et al., 2012*; *Henry et al., 2018*; *Penney et al., 2012*), and another study found that a high dose of rapamycin (3 μm) depleted dopaminergic SVs from presynaptic terminals in the striatum (*Hernandez et al., 2012*), but none of these studies examined SV release itself. Interestingly, several recent studies have uncovered roles for IGF-1 receptor signaling, protein synthesis, and cholesterol biosynthesis in regulating the balance of spontaneous and evoked SV release (*Gazit et al., 2016*; *Scarnati et al., 2018*; *Wasser et al., 2007*). Because these processes are up- and downstream of mTOR, this raises the possibility that mTOR may act as a hub to regulate different modes of SV fusion.

To assess the pre- and postsynaptic function of each mTOR complex in glutamatergic synaptic transmission, we inactivated mTORC1 signaling by conditionally deleting Raptor, or mTORC2 signaling by conditionally deleting Rictor, postmitotically in primary neuron cultures from mouse hippocampus. We then performed morphological and whole-cell patch-clamp analysis of synaptic and membrane properties of glutamatergic neurons. Our results showed that both mTOR complexes were necessary to support normal neuron growth and evoked excitatory synaptic transmission. Despite these similarities, the effects of mTORC1 on evoked EPSCs (eEPSCs) were postsynaptic, via reductions in synapse number, whereas mTORC2 regulated the presynaptic $Ca^{2+}$ dependence of evoked SV release. Furthermore, although the mechanism through which mTORC1 inactivation decreased eEPSCs was postsynaptic, it also increased spontaneous SV release and SV pool replenishment, which are thought to be presynaptic processes. Overall, each mTOR complex affected distinct modes of SV release: mTORC1 inactivation enhanced modes with low rates of SV fusion, such as spontaneous release, and mTORC2 inactivation impaired modes with high rates of SV fusion, such as action potential-evoked release. Thus, via differential activation of these two complexes, the mTOR pathway is ideally poised to respond to external cues and make fine adjustments to glutamatergic synaptic transmission to maintain normal neural network function.

## Results

### Inactivation of mTORC1 or mTORC2 in neurons, via *Raptor* or *Rictor* deletion, causes distinct effects on downstream mTOR signaling

To investigate the effects of mTORC1 or mTORC2 inactivation on neurons, we cultured primary hippocampal neurons isolated from P0-P1 *Raptor*<sup>flox/flox</sup> or *Rictor*<sup>flox/flox</sup> mice on astrocytes isolated from P0-P1 wild-type mice. At the time of plating, we transduced the neurons with adeno-associated viruses (AAVs) expressing either an mCherry-Cre fusion protein or mCherry alone, both driven by the SYN promoter, to generate knockout (Raptor-KO or Rictor-KO) or control (Raptor-Con or Rictor-Con) neurons, respectively. Western blot analyses of protein lysates isolated after 12 days in vitro (DIV) showed significant reductions in both Raptor and Rictor protein levels (*Figure 1—figure*

*supplement 1*), which, given the presence of wild-type astrocytes in the culture, indicated a high efficiency of gene deletion.

To assess the effects of Raptor and Rictor loss on mTORC1 and mTORC2 activity in neurons, we performed quantitative immunofluorescence analysis for phospho-S6 (S240/244) (pS6), an indicator of mTORC1 activity, and phospho-Akt (S473) (pAkt), an indicator of mTORC2 activity. In Raptor-KO neurons, the mean fluorescence intensity of the pS6 signal was decreased relative to Raptor-Con neurons (Con: $1.00 \pm 0.05$, Rap-KO: $0.25 \pm 0.03$, p<0.001; *Figure 1A$_{1-3}$*), verifying a reduction in mTORC1 activity. The pAkt signal was, however, increased in Raptor-KO neurons relative to Raptor-Con neurons (Con: $1.00 \pm 0.05$, Rap-KO: $1.31 \pm 0.06$, p<0.001; *Figure 1B$_{1-3}$*), likely due to release of the negative feedback loop from mTORC1 to insulin receptor signaling (*Hsu et al., 2011*; *O'Reilly et al., 2006*). Rictor-KO neurons showed reduced levels of pAkt immunofluorescence relative to Rictor-Con neurons (Con: $1.00 \pm 0.03$, Ric-KO: $0.56 \pm 0.02$, p<0.001; *Figure 1E$_{1-3}$*), as expected because the S473 residue on Akt is a known target of mTORC2. pS6 levels were also reduced in Rictor-KO neurons compared with those of Rictor-Con neurons (Con: $1.00 \pm 0.05$, Ric-KO: $0.55 \pm 0.03$, p<0.001; *Figure 1F$_{1-3}$*), likely because reduced Akt phosphorylation dampens signaling through the Akt-Tsc-mTORC1 axis. Thus, Cre expression in *Raptor*$^{flox/flox}$ and *Rictor*$^{flox/flox}$ neurons causes loss of Raptor or Rictor protein and distinct biochemical changes consistent with reductions in mTORC1 and mTORC2 activity, respectively.

## Raptor or Rictor loss results in similar effects on neuron morphology and passive membrane properties

In addition to biochemical markers, mTOR signaling is known to regulate cell size (*Edinger and Thompson, 2002*; *Kim et al., 2002*; *Urbanska et al., 2012*); in particular, dramatic increases in cell size occur in most cell types following hyperactivation of the mTOR pathway. To assess alterations in neuron size, we measured neuronal soma area in the cultures following deletion of *Raptor* and *Rictor*. For Raptor-KO and Rictor-KO neurons, soma areas were reduced by almost 20% compared with those of their respective controls (Con: $180 \pm 8$ μm$^2$, Rap-KO: $146 \pm 6$ μm$^2$, p=0.007; Con: $151 \pm 5$ μm$^2$, Ric-KO: $126 \pm 4$ μm$^2$, p<0.001; *Figure 1C and G*).

Previously, mTORC1 and mTORC2 were also shown to regulate dendritic growth (*Urbanska et al., 2012*). Thus, we visualized dendrites in single-neuron cultures, where they can be well-resolved by immunostaining with an antibody against MAP2. We reconstructed the dendritic tree of each neuron and found that total dendritic length was reduced by both mTORC1 inactivation (Con: $1260 \pm 116$ μm, Rap-KO: $747 \pm 70$ μm, p<0.001; *Figure 1D*), and mTORC2 inactivation (Con: $1035 \pm 76$ μm, Ric-KO: $815 \pm 78$ μm, p=0.043; *Figure 1H*). These data verify that a reduction in mTORC1 or mTORC2 activity is sufficient to decrease neuronal soma area and dendritic length, which agrees with two previous studies comparing the effects of mTORC1 and mTORC2 inactivation (*Angliker et al., 2015*; *Urbanska et al., 2012*), and confirm that both mTOR complexes are required for proper neuronal morphology.

The decreases in soma size and dendritic length predict that the passive membrane properties of Raptor-KO and Rictor-KO neurons are altered. To test this, we performed current-clamp analysis of neurons from each group to assess alterations in passive membrane properties and action potential (AP) dynamics (*Table 1*). As may be expected from their decreased soma size, the input resistances of Raptor-KO and Rictor-KO neurons were increased compared with those of their respective controls (Con: $277 \pm 37$ MΩ, Rap-KO: $442 \pm 33$ MΩ, p=0.003; Con: $237 \pm 28$ MΩ, Ric-KO: $329 \pm 37$ MΩ, p=0.033). Also reflective of their reduced soma size, the membrane capacitance (C$_m$) of Raptor-KO neurons was significantly lower than that of Raptor-Con neurons (Con: $138 \pm 8.2$ pF, Rap-KO: $87 \pm 7.4$ pF, p<0.001); however, the effect of Rictor loss on reducing the C$_m$ did not reach statistical significance (Con: $173 \pm 14$ pF, Ric-KO: $135 \pm 10$ pF, p=0.065).

## Deletion of *Raptor* decreases glutamatergic synaptic strength in single-neuron cultures by decreasing quantal size and synapse number

To identify specific roles for mTORC1 and mTORC2 signaling in the regulation of glutamatergic synaptic transmission, we used a single-neuron culture system, which allows for the quantification of multiple parameters of pre- and postsynaptic function in the absence of network compensation and synaptic plasticity. We first performed whole-cell voltage-clamp recordings of glutamatergic neurons

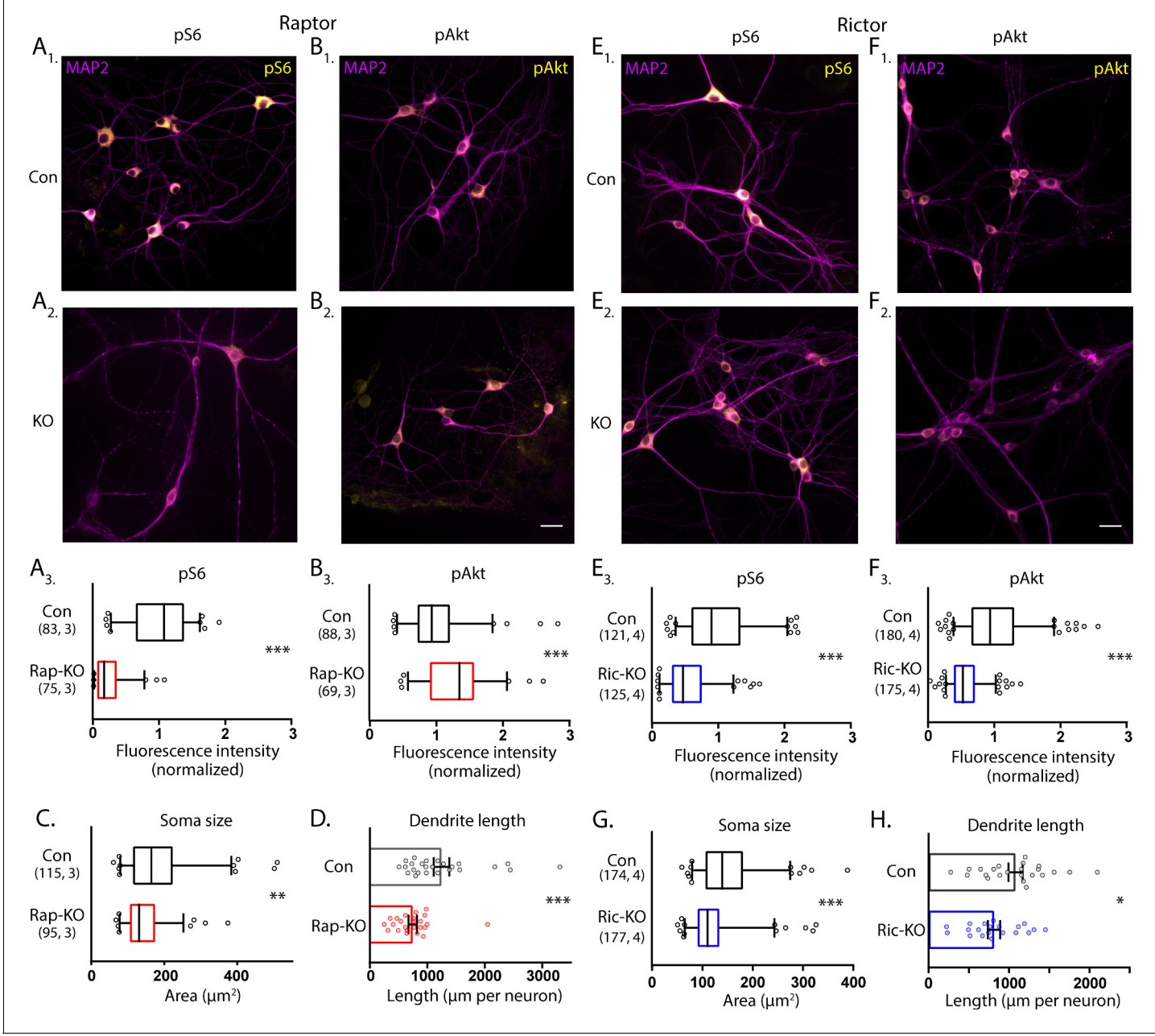

**Figure 1.** Loss of Raptor or Rictor in primary hippocampal neurons alters mTOR signaling and decreases neuron growth. (A) Representative images of Raptor-Con (A₁) and Raptor-KO neurons (A₂) showing the structure of the neurons revealed by MAP2 immunofluorescence (purple) and the immunofluorescence signal from phospho-S6 (pS6, yellow) (A₃) Box plot (median and 95%) showing the fluorescence intensity measurements from the pS6 signal in Raptor-Con and Raptor-KO neurons. (B) Representative images of Raptor-Con (B₁) and Raptor-KO neurons (B₂) showing the structure of the neurons revealed by MAP2 immunofluorescence (purple) and the immunofluorescence signal from phospho-AKT₄₇₃ (pAkt, yellow) Scale bar is 25 μm. (B₃) Box plot (median and 95%) showing the fluorescence intensity measurements from the pAkt signal in Raptor-Con and Raptor-KO neurons. (C) Box plot (median and 95%) showing the area measurements of the somatic compartment in Raptor-Con and Raptor-KO neurons. (D) Dot plot showing the measurements of the total dendritic length in Raptor-Con and Raptor-KO neurons and mean ± s.e.m. Each dot is one neuron sampled from three independent cultures. (E) Representative images of Rictor-Con (E₁) and Rictor-KO neurons (E₂) showing the structure of the neurons revealed by MAP2 immunofluorescence (purple) and the immunofluorescence signal from phospho-S6 (pS6, yellow) (E₃) Box plot (median and 95%) showing the fluorescence intensity measurements from the pS6 signal in Rictor-Con and Rictor-KO neurons. (F) Representative images of Rictor-Con (F₁) and Rictor-KO neurons (F₂) showing the structure of the neurons revealed by MAP2 immunofluorescence (purple) and the immunofluorescence signal from phospho-AKT₄₇₃ (pAkt, yellow) Scale bar is 25 μm. (F₃) Box plot (median and 95%) showing the fluorescence intensity measurements from the pAkt signal in Rictor-Con and Rictor-KO neurons. (G) Box plot (median and 95%) showing the area measurements of the somatic compartment in Rictor-Con and Rictor-KO neurons. (H) Dot plot showing the measurements of the total dendritic length in Rictor-Con and Rictor-KO neurons and mean ± s.e.m.

*Figure 1 continued on next page*

*Figure 1 continued*

Each dot is one neuron sampled from three independent cultures. The numbers underneath the groups indicate the number of neurons analyzed and the number of cultures. *=p < 0.05, **=p < 0.01, ***=p < 0.001, as tested with Generalized Estimating Equations.

The online version of this article includes the following source data and figure supplement(s) for figure 1:

**Source data 1.** Values and statistical analysis of band intensities for Western blot analysis of RAPTOR and RICTOR protein levels.

**Figure supplement 1.** Western blot analysis of Raptor and Rictor levels after Cre-mediated gene deletion.

and evoked APs to examine evoked excitatory postsynaptic currents (eEPSCs) from Raptor-Con and Raptor-KO single neurons. Although the fast component tau of the eEPSC decay was unaltered (Con: $5.65 \pm 0.32$, Rap-KO: $5.67 \pm 0.31$, p=0.45: *Figure 2A$_{1,3}$*), the eEPSC amplitudes were reduced by almost 60% in Raptor-KO neurons relative to those of Raptor-Con neurons (Con: $6.65 \pm 1.38$ nA, Rap-KO: $2.80 \pm 0.57$ nA, p=0.003; *Figure 2A$_{1,2}$*), showing that intact mTORC1 signaling is necessary for normal evoked glutamatergic transmission.

The observed reduction in evoked synaptic strength following Raptor loss could be due to changes in the amplitude of the postsynaptic response to single SV fusion (quantal size) or the number of fusion-competent SVs (readily releasable pool, RRP). First, to test for changes in quantal size, we recorded miniature EPSCs (mEPSCs) and measured their amplitude and decay time constants, properties that are primarily dictated by postsynaptic ionotropic receptor levels and/or activity. Raptor-KO significantly decreased the mEPSC amplitude (Con: $27.8 \pm 1.9$ pA, Rap-KO: $21.5 \pm 1.4$ pA, p=0.006; *Figure 2B$_{1,2}$*), but did not affect the decay time (Con: $3.48 \pm 0.14$ ms, Rap-KO: $3.40 \pm 0.13$ ms, p=0.68; *Figure 2B$_3$*), when compared with those of Raptor-Con neurons. This reduction in quantal size implicates a postsynaptic impairment in Raptor-KO neurons that at least partially accounts for their decrease in evoked glutamatergic release.

Next, we assessed the number of SVs in the RRP following *Raptor* deletion, which can be directly quantified in a single neuron by applying a pulse of hypertonic sucrose (500 mM) to induce the exocytosis of all of a neuron's fusion-competent vesicles (*Rosenmund and Stevens, 1996*). The integral of the transient current during sucrose application represents the total charge contained in the RRP (RRP$_{suc}$), and the total number of vesicles in the RRP can then be calculated by dividing the total charge by the average charge of the miniature events from each neuron. We found that RRP$_{suc}$ in glutamatergic Raptor-KO neurons was decreased by almost 60% compared with that of Raptor-Con

**Table 1.** Summary of the measurements of basic membrane properties.
Measurements are estimated marginal means ± s.e.m. Significance tested with Generalized Estimating Equations.

**Raptor-KO**

|  | Con, n = 13 | Rap-KO, n = 15 | P value | 95% CI of difference |
| --- | --- | --- | --- | --- |
| Resting Potential, mV | $53.3 \pm 2.2$ | $55.9 \pm 2.0$ | 0.38 | −3.4–8.6 |
| Input Resistance, MΩ | $277 \pm 37$ | $442 \pm 33$ | 0.003 | 63–268 |
| Capacitance, pF | $138 \pm 8.2$ | $87 \pm 7.4$ | <0.001 | −73.2 - −27.8 |
| Time constant, ms | $36.8 \pm 2.9$ | $36.2 \pm 2.6$ | 0.89 | −8.6–7.4 |
| AP threshold, mV | $37.1 \pm 2.2$ | $38.0 \pm 2.0$ | 0.76 | −5.1–7.0 |
| AP amplitude, mV | $77.8 \pm 3.5$ | $67.7 \pm 3.1$ | 0.043 | −19.7 - −0.34 |

**Rictor-KO**

|  | Con, n = 15 | Ric-KO, n = 17 | P value | 95% CI of difference |
| --- | --- | --- | --- | --- |
| Resting Potential, mV | $53.5 \pm 1.1$ | $52.2 \pm 1.2$ | 0.41 | −4.5–1.8 |
| Input Resistance, MΩ | $237 \pm 28$ | $329 \pm 37$ | 0.033 | 6–176 |
| Capacitance, pF | $173 \pm 14$ | $135 \pm 10$ | 0.065 | −78.6–2.7 |
| Time constant, ms | $37.4 \pm 5.1$ | $40.3 \pm 5.2$ | 0.65 | −9.6–15.4 |
| AP threshold, mV | $35.0 \pm 1.3$ | $32.3 \pm 1.3$ | 0.14 | −6.5–0.99 |
| AP amplitude, mV | $73.9 \pm 3.2$ | $69.0 \pm 2.99$ | 0.27 | −13.8–4.1 |

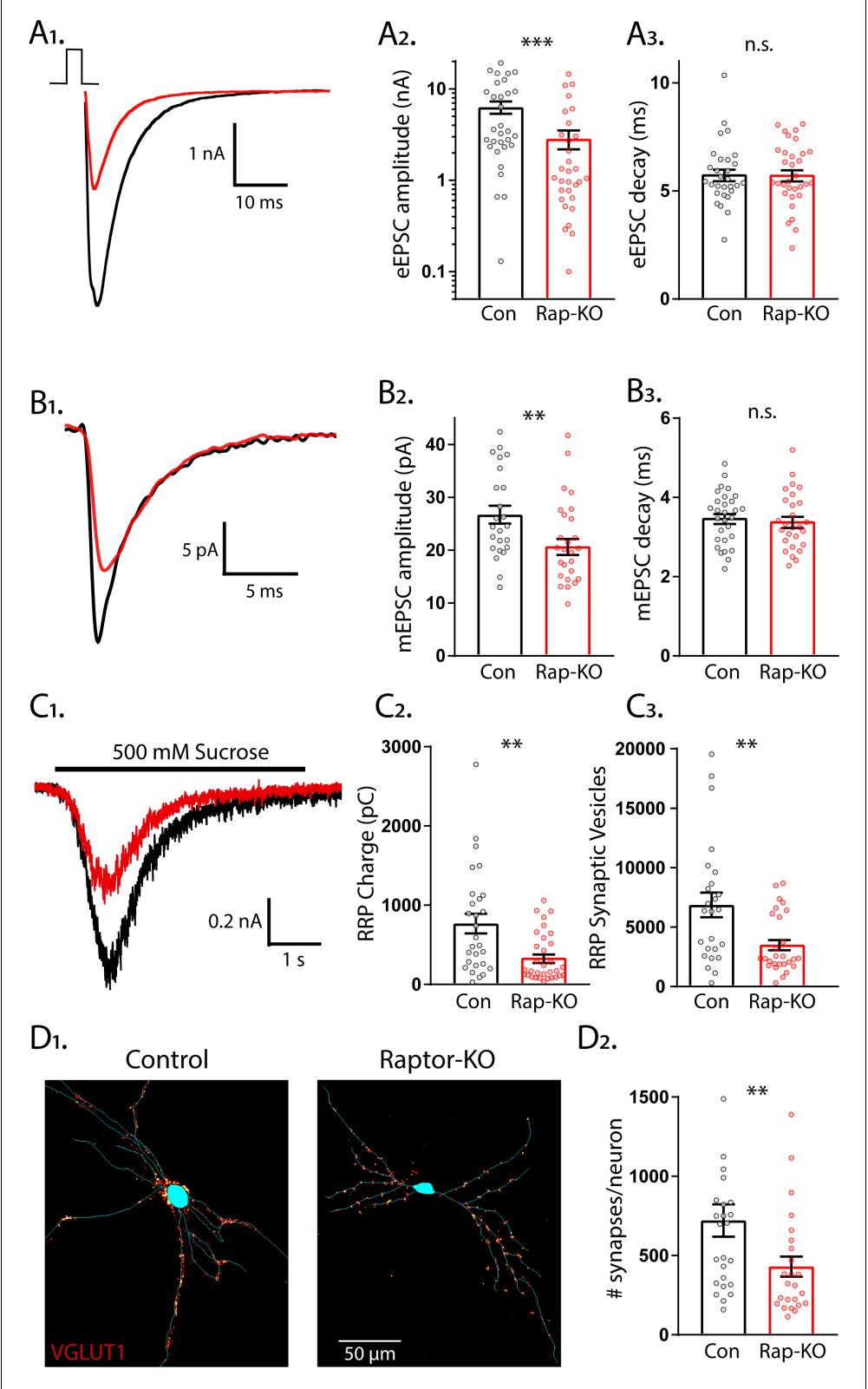

**Figure 2.** Loss of Raptor decreases the strength of evoked excitatory synaptic transmission via changes in quantal size and synapse number. ($A_1$) Example traces of evoked EPSCs (eEPSCs) recorded from single-neuron primary hippocampal cultures of Raptor-Con (black) and Raptor-KO (red) neurons. ($A_2$) Plot showing the values of peak eEPSC amplitudes recorded from Raptor-Con (black) and Raptor-KO (red) neurons on a logarithmic scale. ($A_3$) Plot showing the values of single exponential fits to the fast component of the eEPSC decay recorded from Raptor-Con (black) and Raptor-KO (red)

*Figure 2 continued on next page*

*Figure 2 continued*

neurons on a linear scale. (B$_1$) Example traces of average miniature EPSCs (mEPSCs) recorded from single-neuron primary hippocampal cultures of Raptor-Con (black) and Raptor-KO (red) neurons. (B$_2$) Plot showing the values of mEPSC peak amplitudes recorded from Raptor-Con (black) and Raptor-KO (red) neurons. (B$_3$) Plot showing the distributions of mEPSC decay time constants recorded from Raptor-Con (black) and Raptor-KO (red) neurons. (C$_1$) Example traces of the current response to 500 mM sucrose application recorded from single-neuron primary hippocampal cultures of Raptor-Con (black) and Raptor-KO (red) neurons. The black line indicates the time of sucrose application. (C$_2$) Plot showing the values of the charge contained in the readily releasable pool (RRP) of Raptor-Con (black) and Raptor-KO (red) neurons, as determined by integrating the sucrose response after subtracting the steady state component. (C$_3$) Plot showing the number of vesicles contained in the RRP of Raptor-Con (black) and Raptor-KO (red) neurons, as determined by dividing the RRP charge by the mean mEPSC charge for each neuron. (D$_1$) Representative images showing fluorescence intensity in a red color look up table (LUT) from VGLUT1 immunostaining superimposed on a tracing of the cell body and dendrites from a Raptor-Con (left) and a Raptor-KO (right) neuron. (D$_2$) Plot showing the values of synapse number per neuron for Raptor-Con (black) and Raptor-KO (red) neurons. For all dot plots, each dot represents the mean response from one neuron sampled from three independent cultures and the bars show the estimated marginal means and standard errors. ** indicates a p value of < 0.01, *** indicates p<0.001 and n.s. indicates p>0.05, as tested with Generalized Estimating Equations.

neurons (Con: 767 ± 127 pC, Rap-KO: 337 ± 53 pC, p=0.002; *Figure 2C$_{1,2}$*). As a result, the mean number of SVs contained in the RRP$_{suc}$ of Raptor-KO glutamatergic neurons was reduced by almost 50% relative to that of Raptor-Con neurons (Con: 7145 ± 1415 vesicles, Rap-KO: 3664 ± 718 vesicles p=0.008; *Figure 2C$_3$*).

The observed decrease in the number of SVs in the RRP$_{suc}$ could be due to a decrease in the total number of synapses per neuron or to a decrease in the number of fusion-competent SVs per synapse. To distinguish between these possibilities, we visualized glutamatergic synapses and dendrites by immunostaining with antibodies against VGLUT1 and MAP2. We found that the number of glutamatergic synapses per neuron was decreased by approximately 40% due to Raptor loss (Con: 721 ± 106 synapses/neuron, Rap-KO: 430 ± 65 synapses/neuron, p=0.02; *Figure 2D$_{1,2}$*). However, based on the mean values for the RRP$_{suc}$ and number of synapses, we estimated similar numbers of fusion-competent SVs per synapse in Raptor-Con neurons (9.91 SVs/synapse) and Raptor-KO neurons (8.52 SVs/synapse). Taken together, these data suggest that the reduction in the RRP caused by *Raptor* loss is due to an impairment in synapse formation or maintenance, and not the number of SVs at each synapse.

## Deletion of *Rictor* decreases glutamatergic synaptic strength in single-neuron cultures without affecting quantal size or synapse number

Next, we examined eEPSCs from Rictor-Con and Rictor-KO single neurons. Like Raptor-KO neurons, the eEPSC amplitudes were reduced by almost 60% relative to Rictor-Con neurons (Con: 4.70 ± 0.78 nA, Ric-KO: 1.90 ± 0.57 nA, p=0.001; *Figure 3A$_{1,2}$*), with no effect on the fast component decay time (Con: 5.25 ± 0.25, Ric-KO: 5.53 ± 0.24, p=0.55: *Figure 3A$_3$*). Because of this similarity in the effect on the evoked response, and because our immunostaining indicated that loss of *Rictor* decreased mTORC1 activity (*Figure 1E*), we hypothesized that the physiological mechanisms would be shared (i.e. quantal size and synapse number reductions). Instead, we found that mTORC2 inhibition via *Rictor* loss did not significantly affect the mEPSC amplitude (Con: 19.9 ± 1.1 pA, Ric-KO: 17.7 ± 0.9 pA, p=0.11; *Figure 3B$_{1,2}$*) or decay time (Con: 3.14 ± 0.11 ms, Ric-KO: 3.10 ± 0.10 ms, p=0.76; *Figure 3B$_3$*) relative to those of Rictor-Con neurons, suggesting that, unlike in Raptor-KO neurons, quantal size alterations do not contribute to the decreased glutamatergic synaptic strength observed following loss of *Rictor*.

We next determined whether alterations in the RRP$_{suc}$ size contributed to the reduced eEPSC amplitude in Rictor-KO neurons. The sucrose-induced charge transfer and the number of SVs in the RRP$_{suc}$ were decreased following *Rictor* deletion, although only by approximately 35% (Con: 474 ± 59 pC, Ric-KO: 304 ± 37 pC, p=0.011; and Con: 6441 ± 857 vesicles, Ric-KO: 4178 ± 535 vesicles, p=0.024; *Figure 3C$_{1-3}$*). In contrast to the reduced synapse number caused by *Raptor* loss, the number of glutamatergic synapses per neuron was not decreased by *Rictor* loss (Con: 764 ± 133 synapses/neuron, Ric-KO: 771 ± 140 synapses/neuron, p=0.97; *Figure 3D$_{1,2}$*), even though total dendritic length was reduced (*Figure 1H*). Based on these numbers, we estimated that the number of fusion-competent SVs per synapse was reduced from 8.43 SVs/synapse in Rictor-Con neurons to 5.42 SVs/synapse in Rictor-KO neurons, which would partially account for the 60% reduction in

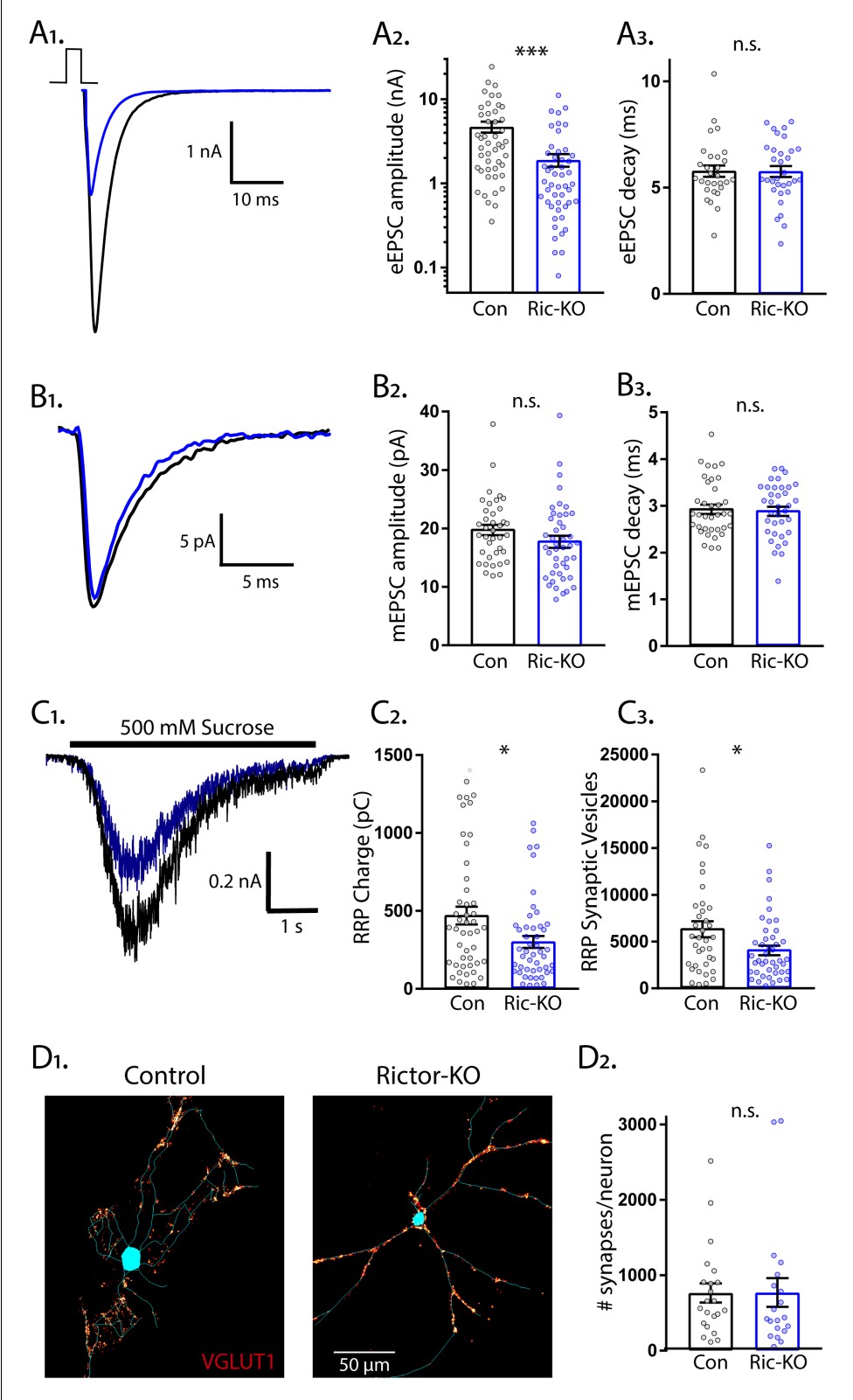

**Figure 3.** Loss of Rictor decreases the strength of evoked excitatory synaptic transmission without altering quantal size or synapse number. (A1) Example traces of eEPSCs recorded from single-neuron primary hippocampal cultures of Rictor-Con (black) and Rictor-KO (blue) neurons. (A2) Plot showing the values of peak eEPSC amplitudes recorded from Rictor-Con (black) and Rictor-KO (blue) neurons on a logarithmic scale. (A3) Plot showing the values of single exponential fits to the fast component of the eEPSC decay recorded from Rictor-Con (black) and Rictor-KO (blue) neurons on a

*Figure 3 continued on next page*

*Figure 3 continued*

linear scale. ($B_1$) Example traces of average mEPSCs recorded from single-neuron primary hippocampal cultures of Rictor-Con (black) and Rictor-KO (blue) neurons. ($B_2$) Plot showing the values of mEPSC peak amplitudes recorded from Rictor-Con (black) and Rictor-KO (blue) neurons. ($B_3$) Plot showing the distributions of mEPSC decay time constants recorded from Rictor-Con (black) and Rictor-KO (blue) neurons. ($C_1$) Example traces of the current response to 500 mM sucrose application recorded from single-neuron primary hippocampal cultures of Rictor-Con (black) and Rictor-KO (blue) neurons. The black line indicates the time of sucrose application. ($C_2$) Plot showing the values of the charge contained in the readily releasable pool (RRP) of Rictor-Con (black) and Rictor-KO (blue) neurons, as determined by integrating the sucrose response after subtracting the steady state component. ($C_3$) Plot showing the number of vesicles contained in the RRP of Rictor-Con (black) and Rictor-KO (blue) neurons, as determined by dividing the RRP charge by the mean mEPSC charge for each neuron. ($D_1$) Representative images showing fluorescence intensity in a red color look up table (LUT) from VGLUT1 immunostaining superimposed on a tracing of the cell body and dendrites from a Rictor-Con (left) and a Rictor-KO (right) neuron. ($D_2$) Plot showing the values of synapse number per neuron neuron for Rictor-Con (black) and Rictor-KO (blue) neurons. For all dot plots, each dot represents the mean response from one neuron sampled from four independent cultures, except the synapse counts, which are from three cultures. The bars show the estimated marginal means and standard errors of the mean. ** indicates a p value of < 0.01, *** indicates p<0.001 and n.s. indicates p>0.05, as tested with Generalized Estimating Equations.

eEPSC amplitude. Thus, although mTORC1 and mTORC2 inactivation both decreased eEPSC strength, our data suggest that the underlying physiological mechanisms are quite different.

## Rictor loss decreases the probability and rate of evoked SV release and increases paired pulse ratios, but Raptor loss does not

For Raptor-KO neurons, the combined decreases in quantal size and the number of SVs in the RRP may be sufficient to account for the magnitude of the decrease in the evoked glutamatergic response. For Rictor-KO neurons, however, the decrease in the evoked glutamatergic response was greater than the decrease in the RRP, indicating that evoked SV release itself may also be impaired. We tested this in three ways. First, we calculated the probability that an SV fuses in response to an AP (vesicular release probability, $P_{vr}$) by dividing the number of vesicles released in response to AP stimulation by the number of vesicles in the $RRP_{suc}$. The $P_{vr}$ was not different between Raptor-KO and Raptor-Con neurons (Con: $0.092 \pm 0.012$, Rap-KO: $0.086 \pm 0.012$, p=0.72; *Figure 4A*). However, Rictor-KO neurons showed a reduced $P_{vr}$ from $0.118 \pm 0.013$, in Rictor-Con neurons, to $0.083 \pm 0.009$ (p=0.010; *Figure 4D*), suggesting that Rictor loss reduces the probability of evoked vesicle fusion. Next, we calculated the peak rate at which SVs were released during the eEPSC by deconvolving the eEPSC with the mean mEPSC shape for each neuron (*Aumann and Parnas, 1991*; *Diamond and Jahr, 1995*; *Schneggenburger and Neher, 2000*). Again, we found no effect of mTORC1 inactivation (Con: $26.0 \pm 3.3$ s$^{-1}$, Rap-KO: $26.1 \pm 3.2$ s$^{-1}$, p=0.98; *Figure 4B$_{1,2}$*), but mTORC2 inactivation decreased the maximum rate of SV release from $22.8 \pm 3.38$ s$^{-1}$ in Rictor-Con neurons, to $11.0 \pm 1.6$ s$^{-1}$ in Rictor-KO neurons (p=0.001; *Figure 4E$_{1,2}$*).

As a third test of SV release changes, we evoked two presynaptic APs in close succession and divided the second postsynaptic response by the first to measure paired-pulse ratios (PPRs) at 25, 50, 100, and 200 ms interstimulus intervals (ISIs) for each neuron group. Generally, in neurons with a lower $P_{vr}$, the second stimulus evokes a larger response than that evoked by the first, resulting in a higher PPR. Conversely, in neurons with a higher $P_{vr}$, the second stimulus evokes a smaller response than that evoked by the first, resulting in a lower $P_{vr}$. In agreement with the $P_{vr}$ measurements, there was no effect of Raptor loss on PPRs at any of the ISIs tested (main effect of group, p=0.35; *Figure 4C$_{1,2}$*). Similarly consistent with the $P_{vr}$ measurements, the Rictor-KO neurons, which had a reduced $P_{vr}$, showed a significant increase in PPRs at all ISIs tested compared with those of Rictor-Con neurons (main effect of group, p<0.001; *Figure 4F$_{1,2}$*). Taken together, these data indicate that the reductions in quantal size and SV number in the RRP of Raptor-KO neurons account for the decreased eEPSC amplitude following inactivation of mTORC1. More importantly, our results strongly suggest that Rictor, but not Raptor, loss leads to presynaptic impairments in evoked vesicle release.

## Postsynaptic loss of Raptor recapitulates the decrease in evoked EPSC amplitude, but Rictor does not

The experiments in the single-neuron cultures indicate that the effects of Raptor loss on eEPSCs are mainly postsynaptic, whereas those of Rictor loss may be presynaptic. In single-neuron cultures,

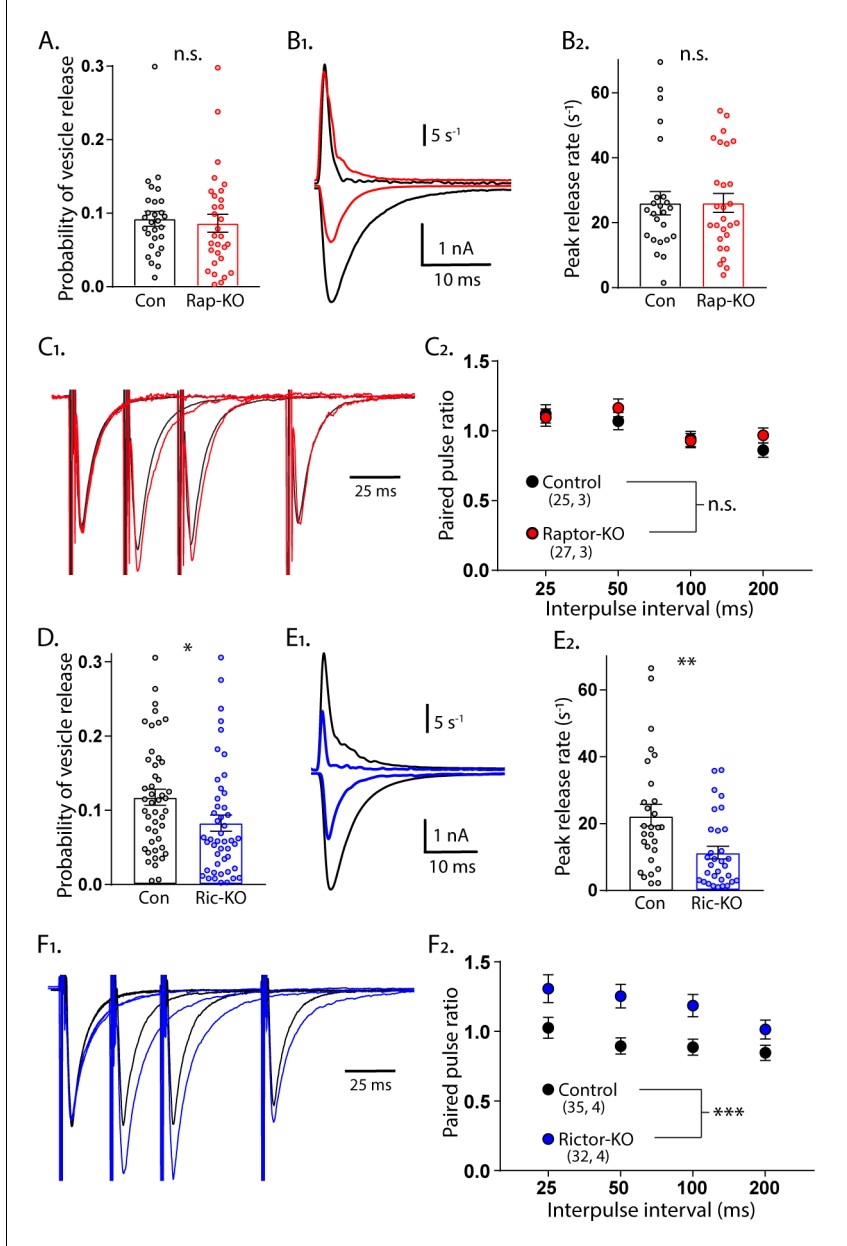

**Figure 4.** Loss of Rictor reduces evoked SV release efficiency but Raptor loss does not. (**A**) Plot showing no difference in the vesicular release probability of Raptor-Con (black) and Raptor-KO (red) neurons. (**B₁**) Example traces showing the rate of SV release (top traces) over their corresponding EPSCs from Raptor-Con (black) and Raptor-KO (red) neurons. (**B₂**) Plot of the peak vesicle release rates of Raptor-Con (black) and Raptor-KO (red) neurons. (**C₁**) Example traces of EPSCs evoked in response to 2 ms depolarizations spaced at 25, 50, and 100 ms. The three sweeps at different intervals are overlayed, as are the responses from Raptor-Con (black) and Raptor-KO (red) neurons. The values are normalized to the peak amplitude of the first EPSC in each sweep. (**C₂**) Summary data showing the estimated marginal means and standard errors for Raptor-Con (black) and Raptor-KO (red) groups at different interpulse intervals. All data summarized in panels (**A-C**) were obtained from three independent cultures. (**D**) Plot showing the decrease in vesicular release probability between Rictor-Con (black) and Rictor-KO (blue) neurons. (**E₁**) Example traces showing the rate of SV release (top traces) over their corresponding EPSCs from Rictor-Con (black) and Rictor-KO (blue) neurons. (**E₂**) Plot of the peak vesicle release rates of Raptor-Con (black) and Raptor-KO (red) neurons. (**F₁**) Example traces of EPSCs evoked in response to 2 ms depolarizations spaced at 25, 50, and 100 ms. The three sweeps at different intervals are overlayed, as are the responses from Rictor-Con (black) and Rictor-KO (blue) neurons. The values are normalized to the peak amplitude of the first EPSC in each sweep. (**F₂**) Summary data showing the estimated marginal means and standard errors for

*Figure 4 continued on next page*

*Figure 4 continued*

Rictor-Con (black) and Raptor-KO (red) groups at different interpulse intervals. All data in panels D-F were obtained from four independent cultures. * indicates p<0.05, ** indicates p<0.01, *** indicates p<0.001, and n.s. = p > 0.05, effect of group tested with Generalized Estimating Equations.

The online version of this article includes the following source data for figure 4:

**Source data 1.** Source data for *Figure 4C$_2$*.

**Source data 2.** Source data for *Figure 4F$_2$*.

however, the pre- and postsynaptic compartments are genetically identical, making it difficult to definitively determine whether a synaptic change is pre- or postsynaptic. To test the hypothesis that mTORC1 regulates eEPSC strength via postsynaptic mechanisms, whereas mTORC2 does not, we recorded postsynaptic responses simultaneously in pairs of control and Raptor-KO or Rictor-KO neurons evoked by optogenetic stimulation of wild-type neurons. This was accomplished by co-culturing three non-overlapping populations of neurons together (wild-type, Cre-expressing, and Chronos-expressing neurons; see Materials and methods) in a traditional mass-neuron culture and then optically stimulating the inputs onto the wild-type and Cre-expressing neurons from a small number of nearby Chronos-expressing neurons. (*Figure 5A*).

We found that the light-evoked (le)EPSC amplitudes evoked onto Raptor-KO neurons were significantly smaller than those onto partner control neurons (Con: 330 ± 63 pA, Rap-KO: 146 ± 28 pA, p<0.001; *Figure 5B$_{1,3}$*). Moreover, the magnitude of this decreased amplitude was equivalent to that initially observed in the eEPSCs recorded from the Raptor-KO single-neuron culture, at almost 60%. Conversely, the leEPSC amplitudes evoked onto Rictor-KO neurons were unaltered relative to those of control neurons (Con: 353 ± 92 pA, Ric-KO: 357 ± 93 pA, p=0.97; *Figure 5C$_{1,3}$*). These data

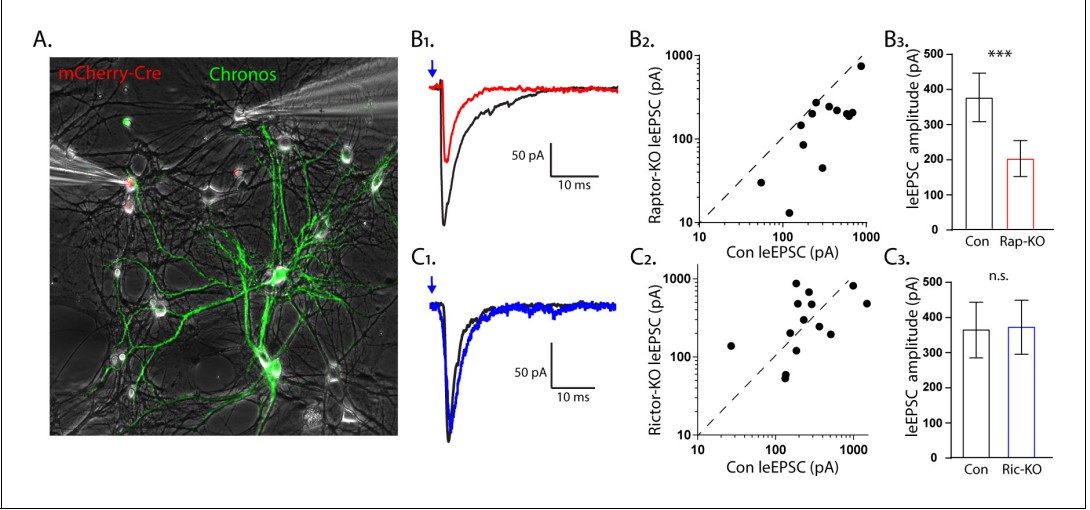

**Figure 5.** The effect of Raptor loss on evoked EPSCs, but not Rictor, is due to postynaptic impairments. (A$_1$) Representative image illustrating the experimental setup. The red fluorescence from the mCherry-Cre and the green fluorescence from the optogenetic protein Chronos fused to GFP are shown overlayed on a phase contrast image showing the patch pipettes attached to one Cre-positive (left pipette) and one Cre-negative (right pipette) neuron. (B$_1$) Example light-evoked (le)EPSCs obtained simultaneously from a control (black) and a Raptor-KO (red) neuron held at −55 mV in response to a 2 ms flash of blue light. (B$_2$) Scatter plot showing the leEPSC responses of neuron pairs to blue light stimulation. The peak amplitude from the control neuron in each pair is represented by the symbol's value on the x-axis, and the peak amplitude of the Raptor-KO neuron is represented by the symbol's value on the y-axis. Pairs in which the KO response is smaller than the control response will be below the dashed line. (B$_3$) Bar graph showing the leEPSC amplitudes (mean ± s.e.m.) of control (black) and Raptor-KO (red) neurons. Data were obtained from three independent cultures. (C$_1$) Example leEPSCs obtained simultaneously from a control (black) and a Rictor-KO (blue) neuron held at −55 mV in response to a 2 ms flash of blue light. (C$_2$) Scatter plot showing the leEPSC responses of neuron pairs to blue light stimulation. The peak amplitude from the control neuron in each pair is represented by the symbol's value on the x-axis, and the peak amplitude of the Rictor-KO neuron is represented by the symbol's value on the y-axis. (C$_3$) Bar graph showing the leEPSC amplitudes (mean ± s.e.m.) of control (black) and Rictor-KO (blue) neurons. Data were obtained from three independent cultures. *** indicates p<0.001 and n.s. = p > 0.05, effect of group tested with Generalized Estimating Equations.

suggest that Raptor, but not Rictor, is required in the postsynaptic neuron to facilitate evoked glutamatergic synaptic transmission.

## mTORC2 regulates presynaptic function by altering Ca²⁺ sensitivity of SV release

The lack of an effect of postsynaptic Rictor loss on eEPSCs, and the alteration in SV release efficiency, suggest that Rictor loss reduces eEPSCs via a presynaptic mechanism. To investigate this, we first asked whether mTORC2 is active at the presynapse by analyzing the colocalization of VGLUT1

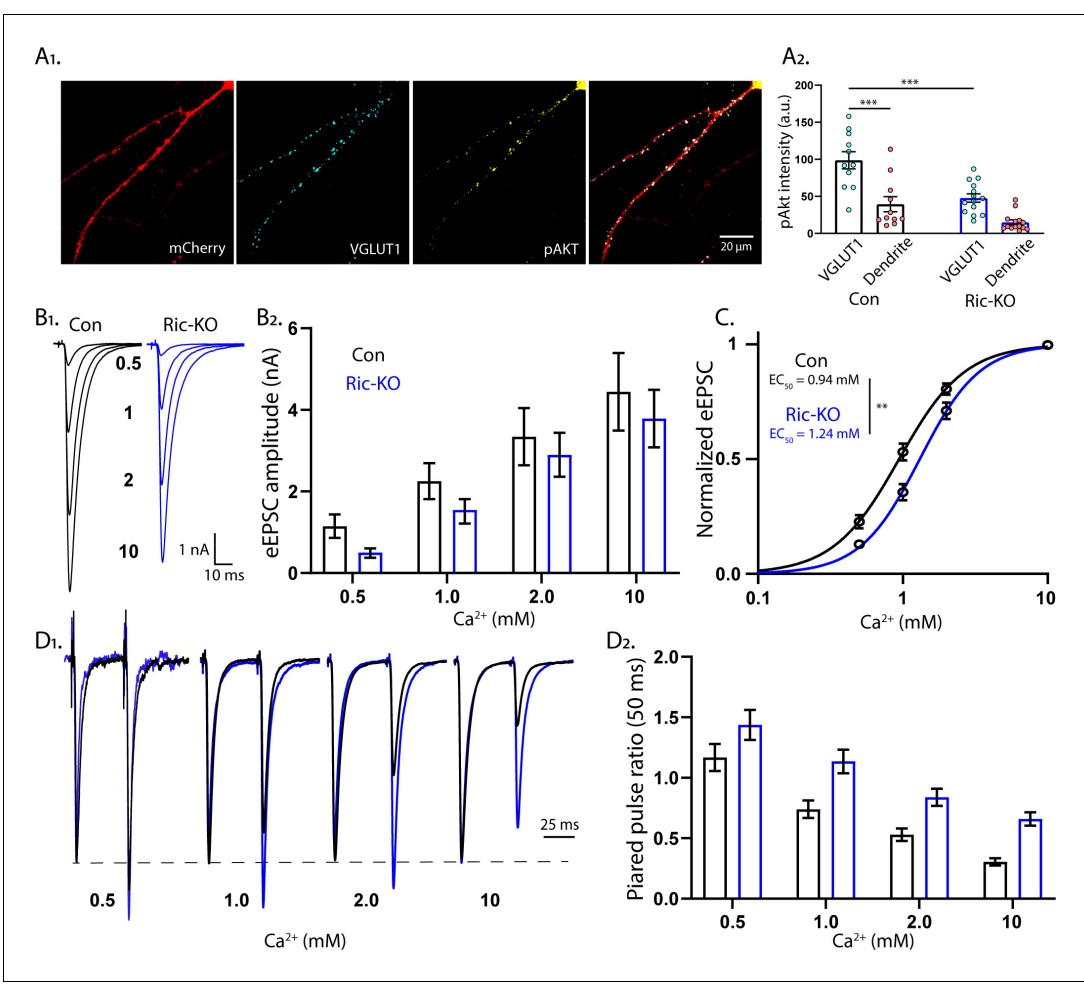

**Figure 6.** mTORC2 acts presynaptically to alter the calcium sensitivity of SV release. (A₁) Representative images showing the colocalization of VGLUT1 (cyan) and pAkt (yellow) immunofluorescent signals on a Rictor-Con neuron (red). (A₂) The pAkt signal colocalized with VGLUT1 is more intense than the signal in neighboring dendrites and is significantly reduced by Rictor loss. Each dot represents the estimated marginal mean value of measurements from one neuron and bars are means ± s.e.m. (B₁) Example traces of eEPSCs in indicated [Ca²⁺]ₑₓₜ from Rictor-Con (black) and Rictor-KO (blue) neurons. (B₂) Estimated marginal means ± s.e.m values for eEPSC amplitudes in indicated [Ca²⁺]ₑₓₜ from Rictor-Con (black) and Rictor-KO (blue) neurons. (C) Linear-log plot of the normalized eEPSC amplitude versus [Ca²⁺]ₑₓₜ for Rictor-Con (black) and Rictor-KO (blue) neurons. Each point is the mean amplitude ± s.e.m. The solid lines are Hill equation fits to the data. (D₁) Example traces of eEPSCs evoked in response to 2 ms depolarizations spaced at 50 ms from Rictor-Con (black) and Rictor-KO (blue) neurons in indicated [Ca²⁺]ₑₓₜ. The traces are normalized to the peak amplitude of the first eEPSC in each [Ca²⁺]ₑₓₜ. (D₂) Summary data showing the paired pulse ratios (estimated marginal means ± s.e.m) for Rictor-Con (black) and Rictor-KO (blue) groups at different [Ca²⁺]ₑₓₜ. All data summarized in panels **A-D** were obtained from three independent cultures. *** indicates p<0.001 as tested with Generalized Estimating Equations.

The online version of this article includes the following source data for figure 6:

**Source data 1.** Source data for *Figure 6B2 and C*.

and pAkt in single-neuron cultures (*Figure 6A*). In Rictor-Con neurons, the intensity of the pAkt signal that colocalized with VGLUT1 was significantly higher than the signal intensity in adjacent dendritic regions (VGLUT1: 102 ± 16, dendrite: 42 ± 7, p<0.0001; *Figure 6A$_{1,2}$*). The VGLUT1-colocalized pAkt signal was significantly reduced in Rictor-KO neurons (Con: 102 ± 16, Ric-KO: 47.7 ± 6, p=0.0002), demonstrating the specificity of the signal. Thus, mTORC2 is active at presynaptic sites in wild-type neurons, and this activity is significantly reduced upon Rictor loss.

Next, we wanted to test whether the reduction in SV release efficiency was due to either a change in the $Ca^{2+}$ cooperativity or potency. To test this, we analyzed the dependence of eEPSC amplitudes on varying external $Ca^{2+}$ concentrations ($[Ca^{2+}]_{ext}$) in Rictor-Con and Rictor-KO neurons (*Figure 6B*). Overall, Rictor-KO neurons had lower eEPSC amplitudes than Rictor-Con neurons (*Figure 6B*), although the reduction appeared stronger at lower $[Ca^{2+}]_{ext}$. We next normalized the response of each neuron to the response in 10 mM $[Ca^{2+}]_{ext}$ and fit a Hill function (*Figure 6C*). The Hill coefficient was not different between the genotypes (Con: 1.92 ± 0.19, Ric-KO: 2.02 ± 0.20, p=0.71), indicating no change in $Ca^{2+}$ cooperativity, but the $EC_{50}$ was significantly higher in Rictor-KO neurons (Con: 0.94 ± 0.05 mM, Ric-KO: 1.24 ± 0.07 mM, p=0.0015; *Figure 6C*), demonstrating that mTORC2 inactivation reduces the apparent sensitivity of SV release to extracellular $Ca^{2+}$. We next determined PPRs at different $[Ca^{2+}]_{ext}$ (50 ms ISI). As before (*Figure 4F*), PPRs were strongly increased in Rictor-KO neurons (*Figure 6D*). Interestingly, in contrast to the reduction in eEPSC amplitudes, which was stronger at lower $[Ca^{2+}]_{ext}$, the effect on PPR appeared stronger at higher $[Ca^{2+}]_{ext}$.

## mTORC1 inactivation accelerates SV pool replenishment

Next, we performed a second analysis of SV release from Raptor-KO and Rictor-KO neurons– application of high frequency stimulus trains– which we analyzed to get a second estimate of the RRP (RRP$_{train}$) and P$_{vr}$, as well as the rate of SV replenishment (*Schneggenburger et al., 1999*). We stimulated neurons at 50 Hz (80 APs), plotted the cumulative EPSC charge versus stimulus number, and fit a line to the last 20 stimuli (*Figure 7A$_1$ and B$_1$*). Using this method, the y-intercept of the linear fit represents the RRP$_{train}$ (*Schneggenburger and Neher, 2000*). In agreement with the RRP$_{suc}$ measurements, Raptor-KO neurons had a significantly reduced RRP$_{train}$ (Con: 202 ± 40 pC, Rap-KO: 43 ± 8 pC, p<0.0001; *Figure 7A$_1$*) and an unchanged P$_{vr}$ (Con: 0.19 ± 0.02, Rap-KO: 0.17 ± 0.02, p=0.59; *Figure 7A$_2$*). Unexpectedly, the rate of SV replenishment, calculated by dividing the slope of the linear fit by the RRP$_{train}$, was significantly increased in Raptor-KO neurons (Con: 0.61 ± 0.14 s$^{-1}$, Rap-KO: 2.73 ± 0.56 s$^{-1}$, p<0.0001; *Figure 7A$_3$*).

In contrast to the RRP$_{suc}$ measurements, the RRP$_{train}$ was indistinguishable between Rictor-Con and Rictor-KO neurons (Con: 199 ± 50 pC, Ric-KO: 228 ± 52 pC, p=0.5; *Figure 7B$_1$*). Despite this, the P$_{vr}$ calculated with this method was again significantly decreased by Rictor loss (Con: 0.16 ± 0.3, Ric-KO: 0.07 ± 0.01, p=0.0002; *Figure 7B$_2$*), whereas the SV replenishment rate was unchanged (Con: 0.63 ± 0.17 s$^{-1}$, Ric-KO: 0.41 ± 0.09 s$^{-1}$, p=0.21; *Figure 7B$_3$*). These data show that the reduction in P$_{vr}$ caused by Rictor loss is robust to two different methods of calculating the RRP size, and that Raptor loss causes an increase in the rate of SV pool replenishment.

To confirm the effect of mTORC1 inactivation on SV replenishment in a different assay, we next measured the rate at which SVs were replenished following the sucrose-induced depletion by analyzing the steady-state component of the current response (*Figure 7C and D*). Again, the rate constant for vesicle replenishment was significantly increased in Raptor-KO neurons compared with that of Raptor-Con neurons (Con: 0.103 ± 0.010 s$^{-1}$, Rap-KO: 0.179 ± 0.017 s$^{-1}$, p<0.001; *Figure 7C$_1$ and 7C$_3$*), but Rictor-KO did not alter the rate of vesicle replenishment (Con: 0.091 ± 0.013 s$^{-1}$, Ric-KO: 0.117 ± 0.017 s$^{-1}$, p=0.32; *Figure 7D$_1$ and 7D$_3$*).

The peak rate of SV release in response to 500 mM sucrose application is thought to reflect the calcium-independent energy barrier for SV fusion (*Basu et al., 2007*). To confirm that the effects of mTORC2 inactivation on SV release rates were calcium dependent, we analyzed the kinetics of the sucrose response by integrating the responses to sucrose for each neuron, converting it to vesicle number and normalizing it to its corresponding RRP, and then finding the maximal slope as a measure for peak release rate (*Basu et al., 2007*). We found that the peak release rate constant induced by sucrose was not different between Raptor-Con and Raptor-KO neurons (Con: 1.54 ± 0.07 s$^{-1}$, Rap-KO: 1.50 ± 0.06 s$^{-1}$, p=0.76; *Figure 7C$_2$*), or Rictor-Con and Rictor-KO neurons (Con: 1.48 ± 0.15 s$^{-1}$, Ric-KO: 1.5 ± 0.17 s$^{-1}$, p=0.76; *Figure 7D$_2$*), confirming that the alteration in SV

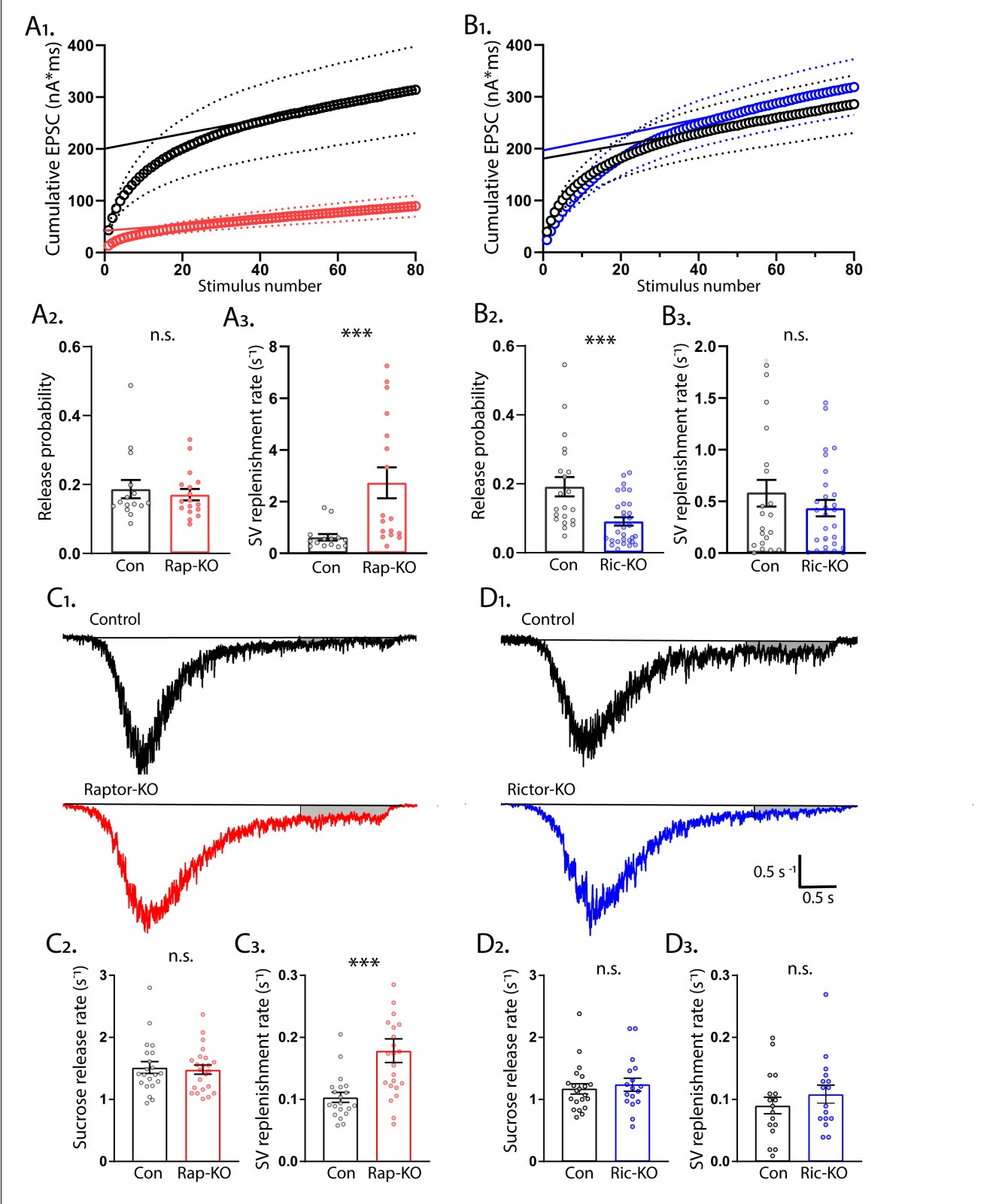

**Figure 7.** mTORC1 inactivation increases SV replenishment rate. (A₁) Cumulative EPSC charge in Raptor-Con (black) and Raptor-KO (red) neurons. Circles are the population means and the dotted lines are the s.e.m. The solid lines are linear fits to the final 20 responses and the y-intercept indicates the RRP_train. (A₂) Release probability calculated using RRP_train is unchanged in Raptor-KO neurons. (A₃) The SV replenishment rate is significantly increased by Raptor loss. (B₁) Cumulative EPSC charge in Rictor-Con (black) and Rictor-KO (blue) neurons. (B₂) Release probability calculated using

*Figure 7 continued on next page*

*Figure 7 continued*

RRP$_{train}$ is significantly decreased in Rictor-KO neurons. ($B_3$) The SV replenishment rate is unaffected by Rictor loss. ($C_1$) Example traces of normalized current responses to 500 mM sucrose application in Raptor-Con (black) and Raptor-KO (red) neurons. The black line shows the pre-sucrose baseline, and the gray shaded area shows the area used to calculate the replenishment rate. ($C_2$) Plot showing the rate constants for sucrose-induced synaptic vesicle (SV) release in Raptor-Con (black) and Raptor-KO (red) neurons. ($C_3$) Plot showing the replenishment rate constants after sucrose-induced SV release in Raptor-Con (black) and Raptor-KO (red) neurons. ($D_1$) Example traces of normalized current responses to 500 mM sucrose application in Rictor-Con (black) and Rictor-KO (blue) neurons. The black line shows the pre-sucrose baseline, and the gray shaded area shows the area used to calculate the replenishment rate. ($D_2$) Plot showing the rate constants for sucrose-induced SV release in Rictor-Con (black) and Rictor-KO (blue) neurons. ($D_3$) Plot showing the replenishment rate constants after sucrose-induced SV release in Rictor-Con (black) and Rictor-KO (blue) neurons. In the dot plots, each dot represents the mean value from one neuron obtained from three independent cultures per genotype, and the bars show the estimated marginal means and s.e.m. ***=p < 0.001, and n.s. = p > 0.05, as tested with Generalized Estimating Equations.

release caused by mTORC inactivation are not due to alterations in the energy barrier for SV fusion, but instead they are due to alterations in the calcium sensitivity of the release process.

## mTORC1 regulates the rate constant for spontaneous SV fusion, but mTORC2 does not

Thus far, we have shown that inactivation of mTORC1 or mTORC2 reduced both evoked EPSC amplitude and RRP size, however, impairments to evoked SV fusion were only observed following mTORC2 inactivation. A decrease in the number of vesicles in the RRP often leads to a decrease in both evoked and spontaneous release, because there is a decreased number of SVs available for fusion, either in response to an action potential or spontaneously (*Schneggenburger and Rosenmund, 2015*). To assess whether spontaneous release is altered by mTORC1 or mTORC2 inactivation, we recorded the frequency of mEPSC events in single-neuron cultures, and then calculated the spontaneous release rate constant (SRR) by dividing the miniature event frequency by the number of SVs in the RRP for each neuron. The SRR is the rate at which an individual SV fuses with the plasma membrane in the absence of stimulation, and the reciprocal of the rate constant is the mean dwell time of an SV in the RRP before it fuses spontaneously. Despite the strong reduction in the RRP caused by loss of synapses in Raptor-KO neurons, the mEPSC frequency was not decreased (Con: 5.77 ± 0.82 Hz, Rap-KO: 6.15 ± 0.87 Hz, p=0.82; *Figure 8A$_{1,2}$*). However, the SRR was significantly increased in Raptor-KO neurons (Con: 0.942 ± 0.18E-3 s$^{-1}$, Rap-KO: 1.751 ± 0.31E-3 s$^{-1}$, p=0.01; *Figure 8A$_3$*), corresponding to mean dwell times of 1062 s and 571 s per SV in Raptor-Con and Raptor-KO neurons, respectively. In Rictor-KO neurons, the mEPSC frequency was decreased relative to Rictor-Con neurons (Con: 3.61 ± 0.89 Hz, Ric-KO: 2.11 ± 0.50 Hz, p=0.012; *Figure 8$_{1,2}$*), but, because of the decrease in the RRP, the SRR was unchanged (Con: 1.14 ± 0.21E-3 s$^{-1}$, Ric-KO: 0.79 ± 0.13E-3 s$^{-1}$, p=0.15; *Figure 8B$_3$*).

Next, we wanted to assess the effect of mTOR inactivation on spontaneous release rates under conditions in which the SRR is elevated. To do this, we stimulated neurons at 10 Hz and then measured spontaneous release in the 10 s following the AP train. Compared to baseline spontaneous release, 10 Hz stimulation caused an increase in spontaneous SV fusion in the 10 s following the end of the train in all groups tested (compare *Figure 8A$_3$ and 8B$_3$* to *Figure 8C$_2$ and 8D$_2$*). Importantly, the mean SRR of Raptor-KO neurons after 10 Hz trains was still higher than that of Raptor-Con neurons (Con: 2.24 ± 0.086E-3 s$^{-1}$, Rap-KO: 5.09 ± 1.07E-3 s$^{-1}$, p=0.001; *Figures 8C1,2*), whereas the SRR following 10 Hz trains in Rictor-KO neurons was not significantly different from that of Rictor-Con neurons (Con: 2.64 ± 0.56E-3 s$^{-1}$, Ric-KO: 1.61 ± 0.33E-3 s$^{-1}$, p=0.096; *Figure 8D$_{1,2}$*). Taken together, these data indicate that, although mTORC1 inactivation reduces the RRP size, the mEPSC frequency is maintained due to an increased rate of spontaneous SV release.

We wanted to ensure that the regulation of SRR by mTORC1 was not unique to the single-neuron culture preparation, but quantification of the SRR in this manner is not possible in mass cultures or brain slice preparations. Instead, we measured mEPSC frequencies after treatment of the neurons with the mTOR inhibitor rapamycin at a time point (12 hr) and concentration (20 nM) at which there should be no change in the number of vesicles in the RRP (*Weston et al., 2012*). Under these assumptions, changes in mEPSC frequency should reflect changes in the SRR due to mTORC1 activity. After 12 hr rapamycin treatment, the mEPSC frequency increased from 4.19 ± 0.60 Hz, in control, to 6.56 ± 0.94 Hz (p=0.016, *Figure 8—figure supplement 1*), a shift similar in magnitude to the one

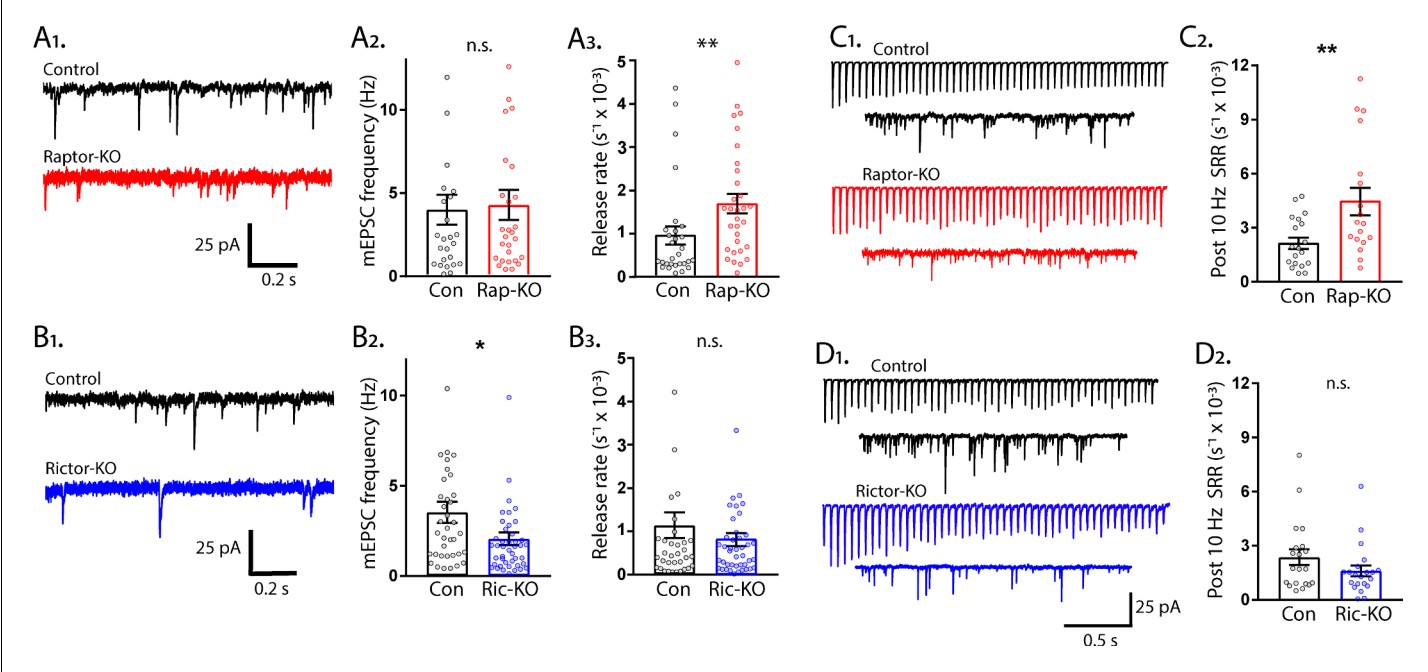

**Figure 8.** The rate constant for spontaneous vesicle fusion is regulated by mTORC1 activity. ($A_1$) Example traces of miniature synaptic currents recorded in single-neuron cultures of Raptor-Con (black) and Raptor-KO (red) neurons. ($A_2$) Plot showing the mEPSC frequencies from Raptor-Con (black) and Raptor-KO (red) neurons. ($A_3$) Plot showing the spontaneous release rate (SRR) of Raptor-Con (black) and Raptor-KO (red) neurons. ($B_1$) Example traces of miniature synaptic currents recorded in single-neuron cultures of Rictor-Con (black) and Rictor-KO (blue) neurons. ($B_2$) Plot showing the mEPSC frequencies from Rictor-Con (black) and Rictor-KO (blue) neurons. ($B_3$) Plot showing the spontaneous release rate (SRR) of Rictor-Con (black) and Rictor-KO (blue) neurons. ($C_1$) Example traces of EPSCs in response to 50 stimulations at 10 Hz (top traces) over traces of spontaneous SV release in the 10 s following the train. Example trace from a Raptor-Con neuron is in black, and a Raptor-KO neuron in red. ($C_2$) Plot showing the rate constants for spontaneous SV release of Raptor-Con (black) and Raptor-KO (red) neurons in the 10 s following the 10 Hz stimulation. ($D_1$) Example traces of EPSCs in response to 50 stimulations at 10 Hz (top traces) over traces of spontaneous SV release in the 10 s following the train. Example trace from a Rictor-Con neuron is in black, and a Rictor-KO neuron in blue. ($D_2$) Plot showing the rate constants for spontaneous SV release of Rictor-Con (black) and Rictor-KO (blue) neurons in the 10 s following the 10 Hz stimulation. In the dot plots, each dot represents the value from one neuron, and the bars show the estimated marginal means and s.e.m. Raptor data was obtained from three independent cultures and Rictor data from four independent cultures. *=p < 0.05, **=p < 0.01, and n.s. = p > 0.05, as tested with Generalized Estimating Equations.

The online version of this article includes the following figure supplement(s) for figure 8:

**Figure supplement 1.** 12 hr Rapamycin treatment increases mEPSC frequency.

caused by *Raptor* deletion, indicating that mTORC1 inhibition increases the rate of spontaneous SV release, regardless of neuron culture conditions.

## mTORC1 and mTORC2 oppositely regulate asynchronous SV fusion

The rates of SV fusion vary over a range of several orders of magnitude, from approximately 0.001 $s^{-1}$ at resting $[Ca^{2+}]_i$ for spontaneous release to approximately 20 $s^{-1}$ during AP-evoked transmitter release at several micromolar of $[Ca^{2+}]_i$ (*Sakaba and Neher, 2001*; *Schneggenburger and Neher, 2000*; *Südhof, 2012*). The data thus far indicate that mTORC1 and mTORC2 regulate the lowest (spontaneous) and highest (AP-evoked) SV fusion rates, respectively. Thus, we next tested how mTORC1 and mTORC2 inactivation affect SV release under conditions in which rate constants are expected to be between these two extremes, asynchronous SV release after a single stimulus and asynchronous fusion during repetitive stimulation.

Asynchronous SV release after a single stimulus in hippocampal neurons accounts for a low percentage of the total transmitter release, but may play important roles in neurotransmission (*Kaeser and Regehr, 2014*), and has been shown to be regulated by SV replenishment (*Otsu et al., 2004*). To quantify the rate constant of asynchronous release, we subtracted the fast component of

evoked release from the total EPSC charge and normalized it by the total RRP charge. Like the SRR, Raptor-KO neurons also showed an increase in the rate of asynchronous SV fusion relative to that of Raptor-Con neurons (Con: $0.070 \pm 0.011$ s$^{-1}$, Rap-KO: $0.122 \pm 0.018$ s$^{-1}$, p=0.002; *Figure 9A$_{1, 2}$*). In contrast to Raptor-KO neurons, but similar to the effect of *Rictor* loss on the peak evoked SV fusion rate, Rictor-KO neurons showed a decrease in the asynchronous SV fusion rate relative to that of Rictor-Con neurons (Con: $0.096 \pm 0.017$ s$^{-1}$, Ric-KO: $0.047 \pm 0.009$ s$^{-1}$, p=0.018; *Figure 9B$_{1, 2}$*).

Asynchronous release during high frequency stimulation can reach high rates of SV fusion, second only to synchronous evoked release. To quantify the asynchronous release rate during 10 Hz stimulation, we subtracted the fast component of evoked release from the total evoked EPSC of the last stimulation of 50 at 10 Hz and normalized this rate by the estimated remaining RRP charge. In this mode of SV release, Raptor-KO neurons did not show an elevated rate of fusion (Con: $1.15 \pm 0.32$ s$^{-1}$, Rap-KO: $1.23 \pm 0.32$ s$^{-1}$, p=0.87; *Figure 9C$_{1, 2}$*), but Rictor-KO neurons did show a significantly lower rate of fusion (Con: $2.21 \pm 0.48$ s$^{-1}$, Ric-KO: $1.19 \pm 0.24$ s$^{-1}$, p=0.037; *Figure 9D$_{1, 2}$*), relative to those of their respective controls.

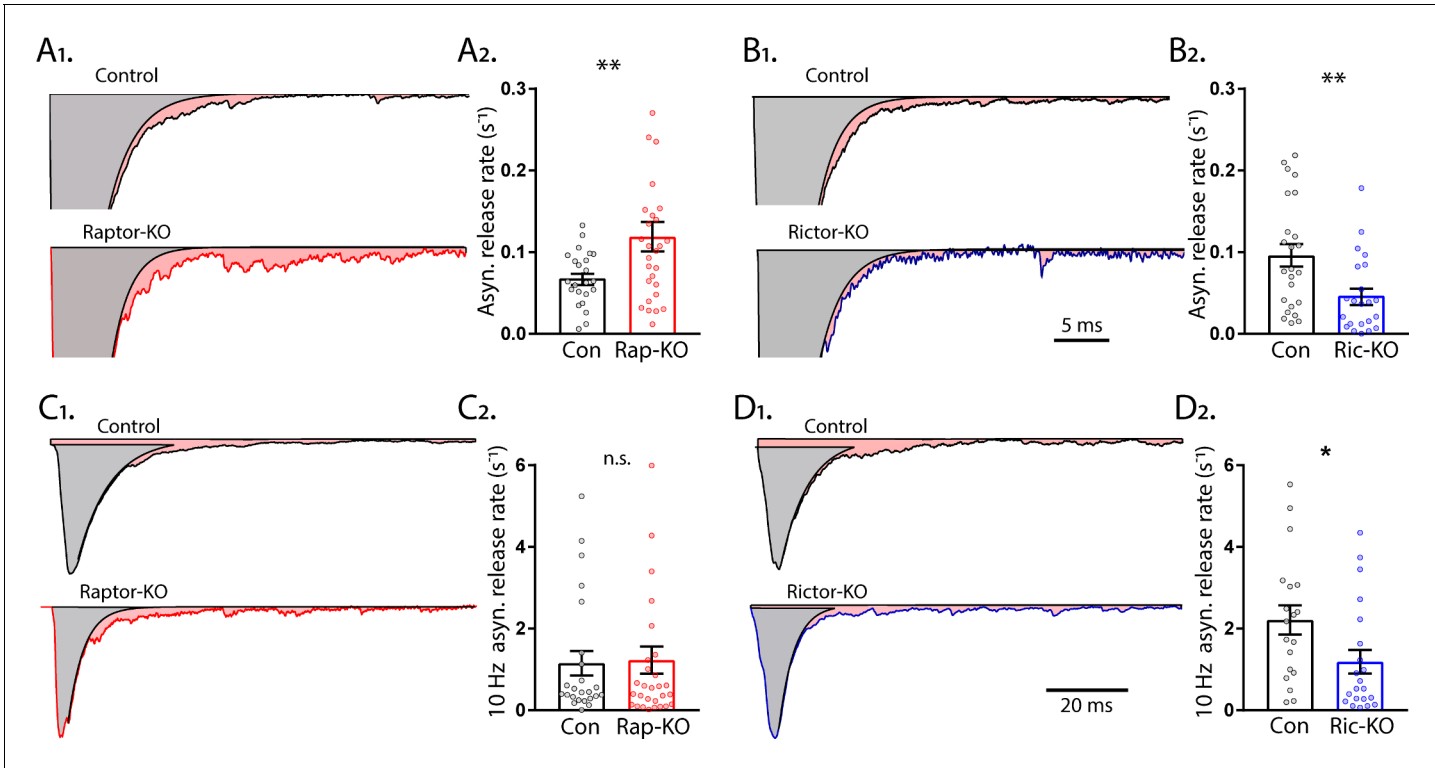

**Figure 9.** mTORC1 and mTORC2 inactivation have opposite effects on asynchronous synaptic vesicle release. (**A$_1$**) Example traces of normalized EPSCs evoked at 0.1 Hz from Raptor-Con (black) and Raptor-KO (red) neurons. The gray shaded area represents the area under the curve of the synchronous component of synaptic vesicle (SV) release, and the pink shaded area represents the asynchronous component. (**A$_2$**) Plot showing the rate constants for asynchronous SV release of Raptor-Con (black) and Raptor-KO (red) neurons. (**B$_1$**) Example traces of normalized EPSCs evoked at 0.1 Hz from Rictor-Con (black) and Rictor-KO (blue) neurons shaded to highlight the synchronous (gray) and asynchronous (pink) components of SV release. (**B$_2$**) Plot showing the rate constants for asynchronous SV release of Rictor-Con (black) and Rictor-KO (blue) neurons. (**C$_1$**) Example traces of normalized EPSCs at the end of a 10 Hz train from Raptor-Con (black) and Raptor-KO (red) neurons. The gray shaded area represents the area under the curve of the synchronous component of SV release and the pink shaded area represents the asynchronous component. (**C$_2$**) Plot showing the rate constants for asynchronous SV release at the end of a 10 Hz train from Raptor-Con (black) and Raptor-KO (red) neurons. (**D$_1$**) Example traces of normalized EPSCs at the end of a 10 Hz train from Rictor-Con (black) and Rictor-KO (blue) neurons shaded to highlight the synchronous (gray) and asynchronous (pink) components of SV release. (**D$_2$**) Plot showing the rate constants for asynchronous SV release at the end of a 10 Hz train from Rictor-Con (black) and Rictor-KO (blue) neurons. In the dot plots, each dot represents the mean value from one neuron obtained from three independent cultures per genotype, and the bars show the estimated marginal means and s.e.m. *=p < 0.05, **=p < 0.01, and n.s. = p > 0.05, as tested with Generalized Estimating Equations.

The online version of this article includes the following figure supplement(s) for figure 9:

**Figure supplement 1.** Summary of SV release rates in Raptor-KO and Rictor-KO neurons.

Finally, to summarize the effects of mTORC1 and mTORC2 inactivation on SV release rates over the range of conditions tested, we plotted the relative changes in rate constants for each of the conditions from lowest to highest rates, plus sucrose (*Figure 9—figure supplement 1*). Taken together, the data indicate that mTORC1 inhibition elevates rate constants for SV fusion under conditions in which the rate is relatively low, but does not affect the rate of fusion when it is high. In contrast, mTORC2 inhibition impairs SV fusion over a wider range of rates, but the effect is more pronounced when rates of SV fusion are high.

## Discussion

Both the mTORC1 and mTORC2 complexes have been shown to regulate important processes such as learning and memory, the response to drugs of abuse, and the development of epilepsy and autism via changes in synaptic strength (*Graber et al., 2013*; *Hou and Klann, 2004*; *Huang et al., 2013*; *Mazei-Robison et al., 2011*; *Stoica et al., 2011*; *Tang et al., 2002*). However, the mechanisms underlying the complex-specific changes in synaptic function are largely unknown. Using genetic mouse models to specifically inactivate mTORC1 or mTORC2, and neuronal culture systems to isolate effects, we show here that both complexes are necessary to support normal neuron growth. Previous studies investigating the specific roles of mTORC1 versus mTORC2 in neurons found that inhibiting either complex via shRNA knockdown of *Raptor* or *Rictor* mRNA in hippocampal neurons (*Urbanska et al., 2012*), or genetic deletion of *Raptor* or *Rictor* in Purkinje neurons (*Angliker et al., 2015*; *Thomanetz et al., 2013*), reduced somatic and dendritic growth. In agreement with these studies, we found reductions in neuronal soma size and dendritic length (*Figure 1*), and corresponding changes in passive membrane properties (*Table 1*), in both Raptor-KO and Rictor-KO neurons, verifying the general role of both mTORC1 and mTORC2 as regulators of neuron growth.

Because of the high level of crosstalk between the two mTOR-containing complexes (*Xie and Proud, 2014*), and because of their similar effects on gross neuronal morphology, it is somewhat surprising that their effects on synaptic transmission are non-overlapping and, in some cases, opposite (*Figure 9—figure supplement 1*). Although inactivation of either complex strongly reduced eEPSC amplitude, we found that the physiological mechanisms underlying these reductions were different, with mTORC1 inhibition reducing eEPSC size via postsynaptic mechanisms and mTORC2 inactivation reducing it through presynaptic mechanisms. Furthermore, we found that mTORC1 inhibition simultaneously increased spontaneous and asynchronous SV release, whereas mTORC2 inhibition decreased evoked and asynchronous SV release. Thus, the roles of mTORC1 and mTORC2 in regulating synaptic transmission are non-overlapping and dissociable from their more general control of neuron growth. Previous studies have shown that synaptic plasticity changes caused by mTOR hyperactivation (PTEN loss) precede large-scale morphological changes (*Sperow et al., 2012*; *Takeuchi et al., 2013*), supporting the idea that synaptic transmission and neuron morphology are independently regulated by mTOR.

Raptor-KO decreased evoked glutamatergic synaptic transmission (*Figure 2*), and postsynaptic Raptor-KO was sufficient to cause this decrease (*Figure 5*). Furthermore, reductions in both the mEPSC amplitude and number of synapses accompanied this decrease (*Figure 2*). Previous studies showed that mTOR inhibition by rapamycin treatment reduces the number of AMPA receptors at the synapse (*Wang et al., 2006*), the number of synapses (*Weston et al., 2012*), and the number of SVs per synapse (*Hernandez et al., 2012*). Accordingly, mTOR hyperactivation increases mEPSC amplitude (*Xiong et al., 2012*), AMPA receptor number, and spine density (*Tang et al., 2014*; *Williams et al., 2015*), and these effects are blocked by rapamycin. Thus, integrating our findings on specific mTORC1 inactivation with these previous findings, several lines of evidence now indicate that mTORC1 acts via a postsynaptic mechanism to bidirectionally regulate evoked glutamatergic synaptic strength.

In contrast to mTORC1 inactivation, mTORC2 inactivation affected presynaptic parameters, including $P_{vr}$, peak evoked SV release rate, and paired pulse ratios (*Figure 4*), and was localized at the presynaptic terminal (*Figure 6*). The $Ca^{2+}$ dose-response curve showed a significant right-shift (*Figure 6*), suggesting that the major mechanism through which mTORC2 inactivation reduces eEPSC strength and presynaptic function is by impairing the $Ca^{2+}$ sensitivity of SV release. Although neurophysiological deficits have been previously reported in Rictor-KO animals, presynaptic function

was not specifically assessed (*Dadalko et al., 2015*; *Huang et al., 2013*; *Thomanetz et al., 2013*; *Zhu et al., 2018*). Thus, our data establish mTORC2 as a potent regulator of presynaptic function and suggest that at least some of the previously reported effects of Rictor loss on synaptic transmission are due to presynaptic deficits.

The different effects following mTORC1 and mTORC2 inactivation on post- and presynaptic function leads to the question of what downstream targets of each complex mediate these changes. Of the molecules downstream of mTORC1, 4E-BP, which is inhibited by active mTORC1, is most strongly linked to regulation of synaptic function. In particular, 4E-BP regulates synaptic transmission via its translational repression of the AMPA receptor subunits GluA1 and GluA2 (*Ran et al., 2013*), and postsynaptic 4E-BP has been shown to play a role in retrograde or trans-synaptic regulation of presynaptic release (*Kauwe et al., 2016*); thus, the effects of Raptor loss could reflect a combination of these effects. Alternatively, a previous study found inhibition of protein synthesis with anisomycin increased the rate of spontaneous SV release and replenishment (*Scarnati et al., 2018*), effects that we also observed in Raptor-KO neurons. It is therefore possible that the regulation of these parameters by mTORC1 is an indirect effect of a reduction in protein synthesis.

Regarding the presynaptic effects of mTORC2, Akt and PKC isoforms are the most well studied substrates. Although Akt has been suggested to play a role in SV endocytosis (*Smillie and Cousin, 2012*), multiple PKC isoforms are targets of mTORC2 (*Ikenoue et al., 2008*; *Thomanetz et al., 2013*). Genetic deletion of PKCβ and PKCγ has been shown to block the $Ca^{2+}$-dependent increase in the RRP and release probability caused by tetanic stimulation at the calyx of Held (*Chu et al., 2014*; *Fioravante et al., 2011*), and pharmacological inhibition of PKC blocks the increase in asynchronous release caused by phorbol esters (*Chang and Mennerick, 2010*). PKC has also been shown to phosphorylate Munc-18 to regulate its interaction with syntaxin (*Fujita et al., 1996*), as well as modulate the actin cytoskeleton in neurons (*Angliker and Rüegg, 2013*). Therefore, the effects of Rictor loss on presynaptic neurotransmission may be caused by lack of PKC activity in these neurons.

Although mTORC1 inactivation decreased evoked strength via postsynaptic mechanisms, it increased the rate of spontaneous and asynchronous release. Because these release rate constants reflect the likelihood of an individual SV to fuse in a given circumstance, they likely reflect a change in the presynaptic terminal. An open question, however, is whether it is reduced mTORC1 activity in the presynapse that causes this change, or whether mTORC1 inactivation in the postsynapse provides a retrograde signal to the presynaptic terminal to alter SV release. In *Drosophila*, 4eBP translationally represses the synaptic protein Complexin to regulate neurotransmitter release at the presynapse (*Mahoney et al., 2016*), however, it is not known if this mechanism is conserved, or if it occurs downstream of the mTOR pathway, in mammals. Another mTORC1 target, SREBP1, regulates cholesterol biosynthesis, and cholesterol depletion has been shown to decrease evoked neurotransmission and enhance spontaneous transmission (*Wasser et al., 2007*), suggesting that this pathway may mediate the effects on presynaptic release in Raptor-KO neurons. However, as mentioned above, there is evidence that postsynaptic mTOR can signal retrogradely to enhance the RRP and presynaptic release in response to a reduction in postsynaptic glutamate receptor activity (*Henry et al., 2012*; *Henry et al., 2018*; *Penney et al., 2012*), providing proof of principle that mTOR can signal across the synapse. In these studies, mTORC1 *activation* signals to increase the RRP, whereas we found that mTORC1 *inactivation* decreases the RRP, and concomitantly increases the spontaneous and asynchronous SV fusion rates. It is possible that mTORC1 regulates additional trans-synaptic signals that regulate spontaneous and asynchronous release, or that the effect on these release modes is merely compensatory downstream of the reduced evoked transmission, but we think the latter scenario is unlikely because 12 hr rapamycin treatment increased spontaneous release, but does not reduce evoked release or synapse number (*Weston et al., 2012*). Furthermore, a recent study found that mTORC1 activity regulates the balance of STX1A to STX1B expression (*Niere et al., 2016*; *Niere and Raab-Graham, 2017*), which, if it occurs at the presynapse, could affect the balance of SV release modes (*Mishima et al., 2014*). Thus, future studies must establish the pre- or postsynaptic locus of the effect of mTORC1 activity on SV release.

One caveat to our findings is that we used the RRP as defined by application of hypertonic sucrose to calculate the rate constants for spontaneous and asynchronous SV fusion. Our analysis assumes that evoked, spontaneous, and asynchronous release all draw from this pool of vesicles. Although there is good evidence to support this assumption (*Ryan et al., 1997*; *Schneggenburger and Rosenmund, 2015*), it is possible that molecularly distinct SV pools are not

all released by sucrose application (*Chamberland and Tóth, 2016*; *Fredj and Burrone, 2009*; *Sara et al., 2005*). If true, this may change our specific conclusion that the rate constant of SV fusion changes to the conclusion that the vesicle pools that support evoked, spontaneous, and asynchronous release are differentially affected by the reduction in synapse number caused by Raptor loss. Thus, although our data cannot conclusively distinguish between these two possibilities, the finding that mTORC1 inactivation enhances asynchronous release after a single stimulation but not after 10 Hz stimulation suggests that it is the SV fusion rate that is affected. We speculate that the differential regulation of SV release modes is due to either the alteration of the sensitivity of SV fusion to calcium (*Nosyreva and Kavalali, 2010*), or an alteration in the influx of calcium under different conditions, which has been shown to differentially regulate spontaneous and evoked release in response to loss of Presenilin 1 (*Pratt et al., 2011*). The fact that we found no changes in the sucrose-evoked SV release rate *(Figure 7)*, which is thought to be calcium-independent (*Rosenmund and Stevens, 1996*), supports the idea that the changes caused by Raptor loss are calcium-dependent.

Several recent studies have highlighted instances in which spontaneous and evoked release are independently modified (*Ramirez and Kavalali, 2011*). It is noteworthy that these examples include blockade of IGF-1 receptors (*Gazit et al., 2016*), inhibition of protein synthesis (*Scarnati et al., 2018*), and reduction of cholesterol levels in neurons (*Wasser et al., 2007*; *Zamir and Charlton, 2006*), as these are all factors that either feed into mTORC1 activity or are modulated by mTORC1 (*Peterson et al., 2011*; *Saxton and Sabatini, 2017*). Because mTORC1 is considered a 'hub' that integrates multiple extra- and intracellular cues to control anabolism in cells (*Kim and Guan, 2019*), it is uniquely positioned to coordinate a synaptic response to metabolic changes. Our findings add to a growing body of literature demonstrating that the metabolic state of neurons can signal to synapses to adjust the balance of spontaneous and evoked release (*Gazit et al., 2016*; *Scarnati et al., 2018*), and identify a novel role for the mTOR signaling network in maintaining this balance. Furthermore, our data broaden the idea of differential regulation of evoked versus spontaneous release by showing that asynchronous release is also affected. Thus, metabolic changes are not mediating a 'competition' or specific ratio between spontaneous and evoked SV release, but instead inducing a shift in the way that synapses respond to different levels of activity, including inactivity.

Variants in at least 10 genes in the mTOR signaling network, including *MTOR*, are known to cause epilepsy, autism, and intellectual disability. Although all of these variants are believed to increase signaling through mTORC1, some have been shown to increase mTORC2 signaling (e.g. *PTEN*, *PIK3CA, and MTOR*), whereas others decrease mTORC2 signaling (e.g. *TSC1*, *TSC2*, and *DEPDC5*). Our data suggest that differential activity levels of the two complexes in disease states would lead to distinct synaptic alterations. Accordingly, previous studies have shown that *Pten* loss and *Tsc1* loss cause different synaptic alterations (*Bateup et al., 2013*; *Williams et al., 2015*). Together, these data suggest that complex specific targeting may be necessary to restore normal synaptic function in neurological diseases involving mTOR hyperactivation. Moreover, future studies are needed to further clarify the contributions of each mTOR complex to these neurological diseases.

## Materials and methods

**Key resources table**

| Reagent type (species) or resource | Designation | Source or reference | Identifiers | Additional information |
|---|---|---|---|---|
| Genetic reagent *Mus musculus* | Raptor-cKO mice | The Jackson Laboratories | Jackson Labs stock: 013188 | |
| Genetic reagent *Mus musculus* | Rictor-cKO mice | The Jackson Laboratories | Jackson Labs stock: 020649 | |
| Strain, strain background *Mus musculus* | C57BL/6J | The Jackson Laboratories | Jackson Labs stock: 000664 | |
| Recombinant DNA reagent | AAV8-SYN-mCherry-Cre | UNC Vector Core | | |
| Recombinant DNA reagent | AAV8-SYN-mCherry | UNC Vector Core | | |

*Continued on next page*

*Continued*

| Reagent type (species) or resource | Designation | Source or reference | Identifiers | Additional information |
|---|---|---|---|---|
| Recombinant DNA reagent | AAV9-Syn-Chronos-GFP | UNC Vector Core | | |
| Antibody | Rabbit monoclonal anti-Raptor | Cell Signaling | Cat #2280 RRID:AB_561245 | (1:1000) |
| Antibody | Rabbit monoclonal anti-Rictor | Cell Signaling | Cat #2114 RRID:AB_2179963 | (1:1000) |
| Antibody | Mouse monoclonal anti-MAP2 | Synaptic Systems | Cat #188 011 | (1:1000) |
| Antibody | rabbit polyclonal anti-VGLUT1 | Synaptic Systems | Cat #135 302 | (1:5000) |
| Antibody | guinea pig polyclonal anti-VGLUT1 | Synaptic Systems | Cat #135 304 | (1:5000) |
| Antibody | Rabbit polyclonal anti-pAkt (S473) | Cell Signaling | Cat # 9271 RRID:AB_2315049 | (1:1000) |
| Antibody | Rabbit monoclonal anti-pS6 (S240/244) | Cell Signaling | Cat # 5364 RRID:AB10694233 | (1:1000) |
| Chemical compound, drug | rapamycin | Cayman | Item # 13346 | |
| Chemical compound, drug | Kynurenic acid | Tocris | Cat. #0223 | |
| Chemical compound, drug | Bicuculline Methiodide | Tocris | Cat. #2503 | |
| Chemical compound, drug | Tetrodotoxin | Enzo | BML-NA120-0001 | |
| Software, algorithm | pClamp | Molecular devices | RRID:SCR_011323 | |
| Software, algorithm | Axograph X | Axograph | RRID:SCR_014284 | |
| Software, algorithm | SPSS | SPSS | RRID:SCR_002865 | |
| Software, algorithm | Prism | Graphpad | RRID:SCR_002798 | |
| Software, algorithm | FIJI | NIH | RRID:SCR_002285 | |
| Software, algorithm | Intellicount | PMID:29218324 | | |
| Software, algorithm | Matlab | Mathworks | RRID:SCR_001622 | |

## Mice and cell culture

Animal housing and use were in compliance with the National Institutes of Health (NIH) Guidelines for the Care and Use of Laboratory Animals and were approved by the Institutional Animal Care and Use Committee at the University of Vermont. Experiments used B6.Cg-*Rptor*^tm1.1Dmsa/J mice that possess loxP sites on either side of exon 6 of the *Raptor* gene (Jackson Labs stock: 013188), and *Rictor*^tm1.1Klg/SjmJ mice that possess loxP sites on either side of exon 11 of the *Rictor* gene (Jackson Labs stock: 020649). Both lines were maintained as homozygous for the floxed allele. Experiments that involved treatment of wild-type neuronal cultures with drugs used C57BL/6J mice (Jackson Labs stock: 000664).

Conventional and single-neuron primary cultures were grown on astrocytes derived from wild-type C57BL/6J mice (Jackson Labs stock: 000664). Cortices were dissected from postnatal day 0–1 (P0-P1) mice of either sex and placed in 0.05% trypsin-EDTA (Gibco) for 15 min at 37°C in a Thermomixer (Eppendorf) with gentle agitation (800 rpm). Then, the cortices were mechanically dissociated with a 1 mL pipette tip and the cells were plated into T-75 flasks containing astrocyte media [DMEM media supplemented with glutamine (Gibco) and MITO+ Serum Extender (Corning). After the astrocytes reached confluency, they were washed with PBS (Gibco) and incubated for 5 min in 0.05% trypsin-EDTA at 37°C, and then resuspended in astrocyte media. For conventional cultures, the astrocytes were added to 6-well plates containing 25 mm coverslips precoated with coating mixture [0.7 mg/ml collagen I (Corning) and 0.1 mg/ml poly-D-lysine (Sigma) in 10 mM acetic acid]. For

single-neuron cultures (*Burgalossi et al., 2012*), the astrocytes were added to 6-well plates containing 25 mm agarose-coated coverslips stamped with coating mixture using a custom-built stamp to achieve uniformly sized, astrocyte microislands (200 µm diameter).

For the primary neuron culture, the hippocampi from P0-P1 mice of both sexes were dissected in cold HBSS (Gibco). The hippocampi were then digested with papain (Worthington) for 60–75 min and treated with inactivating solution (Worthington) for 10 min, both while shaking at 800 rpm at 37° C in a Thermomixer. The neurons were then mechanically dissociated and counted. For single-neuron cultures, 2000–3000 neurons/well were added to 6-well plates in NBA plus [Neurobasal-A medium (Gibco) supplemented with Glutamax (Gibco) and B27 (Invitrogen)], each well containing a 25 mm coverslip with astrocyte microislands. For conventional cultures for immunofluorescence analysis, 150,000 neurons/well were added to 6-well plates in NBA plus, each well containing a 25 mm coverslip with a confluent layer of astrocytes. For Western blot analysis, 250,000–300,000 neurons/well were added. After plating, approximately $4 \times 10^{10}$ genome copies (GC) of either AAV8-SYN-mCherry-Cre or AAV8-SYN-mCherry virus (UNC Vector Core) was added to each well. For the treatment of WT neurons with rapamycin experiment, rapamycin (Cayman Chemical) was dissolved in DMSO at a concentration of 20 µM and then added to cell culture media at a 1:1000 dilution for 12–16 hr prior to electrophysiology experiments to achieve a final concentration of 20 nM. Control neurons were treated with an equal amount of DMSO alone.

For experiments using paired recording and optogenetic excitation, the neuron suspensions after hippocampal dissociation were split into three tubes of 300 µl each. To one of these tubes, $6 \times 10^{10}$ GC of AAV8-SYN-mCherry-Cre was added, and to another, $6 \times 10^{10}$ GC of AAV9-Syn-Chronos-GFP (*Klapoetke et al., 2014*) was added. The virus was left on for 3 hr while the neuron suspensions were gently shaken (500 rpm) at 37°C in a Thermomixer. After 3 hr, the neurons were centrifuged three times at 1500 rpm for 5 min on a benchtop centrifuge and resuspended in fresh Neurobasal-A medium each time. After the third resuspension, the neurons were counted. From each of the three tubes, 50,000 neurons were added to each well of a 6-well plate containing NBA plus to generate a network containing Control, Cre-expressing, and Chronos-expressing neurons in non-overlapping neuronal populations.

## Protein extraction and western blot

Briefly, each well of a 6-well tissue culture plate was scraped into a buffer containing 8M Urea and 1% CHAPS buffer then subject to sonication. The resulting cell lysate was clarified by centrifugation and the protein-containing supernatant was used for analysis. Twenty micrograms of protein were run in each lane of a 4–12% NuPAGE bis-tris gradient gel (Invitrogen NP0335BOX). Following transfer, the PVDF membrane was blocked in milk and probed with antibodies raised against RAPTOR (Cell Signaling 2280), RICTOR (Cell Signaling, 2114) TUBULIN (ProteinTech 66240) or GAPDH (ProteinTech 60004). The appropriate HRP-conjugated antibody was then applied and imaged following application of enhanced chemiluminescent substrate (Pierce 32106). The pixel intensity of each band was quantified in Image J. RAPTOR and RICTOR band intensities were normalized to the averaged intensity of GAPDH and TUBULIN.

## Immunocytochemistry

Neurons were rinsed three times with PBS, fixed with 4% PFA for 30 min, and then washed with PBS three times. Neurons were then placed in blocking solution (10% NGS, 0.1% Triton X-100, and PBS) at room temperature for 1 hr. The following primary antibodies in blocking solution were then applied to the neurons at 4°C overnight: MAP2 (mouse monoclonal, 1:1000 dilution, Synaptic Systems, Cat# 188 011, RRID:AB_2147096), phospho-S6 Ribosomal Protein Ser240/244 (rabbit monoclonal, 1:1000 dilution, Cell Signaling Technology, Cat# 5364, RRID:AB_10694233), phospho-AKT Ser473 (rabbit monoclonal, 1:1000 dilution, Cell Signaling Technology, Cat# 4060, RRID:AB_2315049), VGLUT1 (rabbit polyclonal, 1:5000 dilution, Synaptic Systems, Cat # 135 302, RRID:AB_887877), or VGLUT1 (guinea pig polyclonal, 1:5000 dilution, Synaptic Systems, Cat # 135 304). Following primary antibody application, cells were washed three times in PBS and then incubated in the following Alexa Fluor secondary antibodies (Invitrogen/Molecular Probes) for 1 hr at room temperature: goat anti-mouse 488 (1:1000, Cat # A-11017, RRID:AB_143160) and goat anti-rabbit 647

(1:1000, Cat# A-21244, RRID:AB_141663). Cells were then mounted to slides with Prolong Gold Antifade (Life Technologies) and allowed to cure for 24 hr.

Images (1024 × 1024 pixels) for pS6 and pAKT expression analysis were obtained using a DeltaVision Restoration Microscopy System (Applied Precision/GE Life Sciences) with an inverted Olympus IX70 microscope with a 20 × oil objective, SoftWoRx software, and a CoolSNAP-HQ charge-coupled device digital camera (Photometrics). Image exposure times and settings were kept the same between groups in a culture and were optimized to ensure that there were no saturated pixels. Images were acquired in stacks of 8–12 planes at 0.5 μm depth intervals and then deconvolved. Stacks were processed using Fiji software (*Schindelin et al., 2012*) to create maximum intensity projections. Image background was subtracted using the rolling ball method with a radius of 100 μm. To analyze levels of mTOR effectors, regions of interest (ROIs) were drawn around the cell body using the MAP2 channel, and then the mean fluorescence intensity and cell body area were measured for pS6 and pAKT for each neuron imaged. Because the absolute values of the fluorescence intensity varied between cultures, the values were normalized to the mean value of the control neurons for each culture.

For dendritic length, glutamatergic terminal number analysis, and pAkt-VGLUT1 colocalization, primary neuron cultures on astrocyte microislands were generated and fixed as described above. Images (1024 × 1024 pixels) were obtained using a C2 confocal microscopy system (Nikon) with a 40x oil objective. Images were acquired using equal exposure times between groups in stacks of 4–6 images at 2.0 μm depth intervals and then randomized and renamed to blind analysis. Maximum intensity projections were created using Fiji software. Total dendritic length was obtained by tracing MAP2 expression using the NeuronJ plugin (*Meijering et al., 2004*). VGLUT puncta number were calculated using Intellicount software (*Fantuzzo et al., 2017*).

## Electrophysiology

Whole-cell recordings were performed with patch-clamp amplifiers (MultiClamp 700B amplifier; Molecular Devices) under the control of Clampex 10.3 or 10.5 (Molecular Devices, pClamp, RRID: SCR_011323). Data were acquired at 10 kHz and low-pass filtered at 6 kHz. The series resistance was compensated at 70%, and only cells with series resistances maintained at less than 15 MΩ and stable holding currents of less than 400 pA were analyzed. The pipette resistance was between 2 and 4 MΩ. Standard extracellular solution contained the following (in mM): 140 NaCl, 2.4 KCl, 10 HEPES, 10 glucose, 4 $MgCl_2$, and 2 $CaCl_2$ (pH 7.3, 305 mOsm). Internal solution contained the following: 136 mM K-gluconate, 17.8 mM HEPES, 1 mM EGTA, 0.6 mM $MgCl_2$, 4 mM ATP, 0.3 mM GTP, 12 mM creatine phosphate, and 50 U/ml phosphocreatine kinase. All experiments were performed at room temperature (22–23°C). Whole-cell recordings were performed on neurons from control and mutant groups in parallel on the same day (day 12–14 in vitro) and in two of the cultures from each genotype the experimenter was blinded to treatment condition.

For voltage-clamp experiments, neurons were held at −70 mV unless noted. Action potential (AP)-evoked EPSCs were triggered by a 2 ms somatic depolarization to 0 mV. The shape of the evoked response and the effect of receptor antagonists [3 mM kynurenic acid (KYN, Tocris Bioscience) or 20 μM bicuculline (BIC, Tocris Bioscience)] were analyzed to verify the glutamatergic or GABAergic identities of the neurons. Neurons were stimulated at 0.2 Hz in standard external solution to measure basal-evoked synaptic responses. Electrophysiology data were analyzed offline with AxoGraph X software (AxoGraph Scientific, RRID:SCR_014284). To determine the number of releasable SVs onto each neuron, we measured the charge transfer of the transient synaptic current induced by a 5 s application of hypertonic sucrose solution directly onto the neuron and then divided the sucrose charge by the charge of the average miniature event onto the same neuron (*Rosenmund and Stevens, 1996*).

For current-clamp experiments, the resting membrane potential was measured and then current was injected to achieve a resting membrane potential of −70 mV. KYN was applied to block synaptic responses. Input resistance and membrane time constant were calculated from the steady state and charging transient, respectively, of voltage responses to 0.5 s, 20 pA hyperpolarizing current steps. Membrane capacitance was calculated by dividing the time constant by the input resistance. AP were evoked with 0.5 s, 20 pA depolarizing current steps. AP threshold was defined as the membrane potential at the inflection point of the rising phase of the AP. AP amplitude was defined as

the difference in membrane potential between the AP peak and threshold. The membrane potential values were not corrected for the liquid junction potential.

For optogenetic stimulation experiments, pairs of Cre-expressing and control neurons were voltage clamped and held at −60 mV, which was the empirically determined reversal potential for GABAergic responses. Light pulses (5 ms at 5% power) were applied via a X-Cite 120 LED (Excelitas) triggered by a TTL output from the digitizer and controlled by Clampex. The light was passed through a Brightline 4050B filter (Semrock) and the 20x objective of the microscope to activate the afferents of a small number of Chronos-expressing neurons onto the recorded neurons.

For eEPSC measurements in varying $Ca^{2+}$ concentrations, EPSCs from hippocampal neurons were measured in extracellular solution containing 0.5 mM, 2 mM, 4 mM, or 10 mM $CaCl_2$ and 1 mM $MgCl_2$ in a randomized order. Responses were also measured in standard external solution (2 mM $CaCl_2$, 4 mM $MgCl_2$) between treatments and cells with a greater than 20% shift in the standard solution ePESC amplitude were not used. The responses were normalized to the response in 10 mM $CaCl_2$ and fit with the Hill equation in Prism 8 (GraphPad Prism, RRID:SCR_002798) to create the dose-response curve and obtain the $EC_{50}$ and Hill coefficient.

## Miniature event detection

Miniature synaptic potentials were recorded for 70–90 s in 500 nM tetrodotoxin (TTX, Enzo Life Sciences) to block AP-evoked release. Data were filtered at 1 kHz and analyzed using template-based miniature event detection algorithms implemented in the AxoGraph X. The threshold for detection was set at three times the baseline SD from a template of 0.5 ms rise time and 3 ms decay. For each neuron, 3 mM KYN was applied as a negative control to detect false positive events. If the frequency of false positives exceeded 0.25 the frequency of total positives, the neuron was discarded. If rate was lower than 0.25, the amplitude and frequency of false positives were subtracted from the total to obtain the rate and frequency of true positives.

## Synaptic vesicle release rate analysis

The rate constant for vesicle fusion (k) was calculated for each neuron and each mode of vesicle release with the first order reaction equation r = k [A], where r = the observed vesicle release rate (SVs/s) and A = the number of SVs in the RRP. For spontaneous release, the observed vesicle release rate was the mEPSC frequency. For the peak rate of evoked SV release, at least 10 EPSCs were collected per neuron, baselined to the 5 ms period immediately preceding the stimulation, filtered at 1 kHz and deconvolved with the waveform of the mean mEPSC from that neuron using a custom algorithm implemented in Axograph X to give the SV release rate waveform. The deconvolved EPSC waveform was then integrated and the maximum slope over a 1 ms time bin was considered the peak rate of SV release. For the spontaneous release rate after 10 Hz stimulation, the vesicle release rate was the mean mEPSC frequency over 10 s beginning 100 ms after the last stimulation in the train. For asynchronous release, the vesicle release rate was calculated by fitting a single exponential to the fast component of the EPSC decay, subtracting the fast component from the total, and then dividing the charge transfer of the remaining response by the charge of the average mEPSC for each neuron. For asynchronous release during 10 Hz stimulation, the vesicle release rate was calculated by baselining the first EPSC in the train, fitting a single exponential to the fast component of the EPSC decay of the last EPSC in the train, subtracting the fast component from the total, and then dividing the charge transfer of the remaining response by the charge of the average mEPSC for each neuron (*Chang and Mennerick, 2010*; *Otsu et al., 2004*). To account for depletion of the pool during the train, A was estimated by multiplying the number of SVs in the RRP times the ratio of the charge of the last EPSC to the charge of the first EPSC in the train. Although we note that this may underestimate the amount of depletion due to an increase in release probability during the train.

## Experimental design and statistical analysis

KaleidaGraph 4.5 (Synergy Software) and Prism 7 (GraphPad Prism, RRID:SCR_002798) were used to create graphs. To test for statistical significance, we used generalized estimating equations (GEE) in SPSS (24.0 Chicago, III (IBM, RRID:SCR_002865), which allows for within-subject correlations and the specification of the most appropriate distribution for the data. All data distributions were assessed with the Shapiro-Wilk test. Datasets that were significantly different from the normal distribution

(p<0.05) were fit with models using the gamma distribution and a log link. Normal datasets were fit with models using a linear distribution and identity link. We used the model-based estimator for the covariance matrix and an exchangeable structure for the working correlation matrix. Goodness of fit was determined using the corrected quasi likelihood under independence model criterion and by the visual assessment of residuals. Because neurons and animals from the same culture are not independent measurements, culture was used as the subject variable, and animals and neurons were considered within-subject measurements. All values reported in the text are estimated marginal means + / - standard error. Sample sizes for microisland culture analysis were estimated based on our previous results from the effects of the mTOR inhibitor rapamycin (*Weston et al., 2012*). To determine our sample size for the experiments using paired recording and optogenetic excitation, we performed a power analysis in G*Power 3.1 (*Faul et al., 2009*) based on the effect size of the EPSC amplitude measurements in the single-neuron cultures and calculated the number of pairs we needed to record from to detect a difference with 80% power at $\alpha = 0.05$ in each group if the effect on EPSC amplitude were purely postsynaptic.

## Acknowledgements

This work was supported by NIH/NINDS grants NS087095 and NS110945, and the COBRE Neuroscience award P30 103498. We thank Todd Clason, and the Molecular and Imaging Cores, at the University of Vermont. Thanks to John Clements for providing the deconvolution algorithm used to estimate the rate of synaptic vesicle release.

## Additional information

### Funding

| Funder | Grant reference number | Author |
| --- | --- | --- |
| National Institute of Neurological Disorders and Stroke | R00NS087095 | Matthew C Weston |
| National Institute of Neurological Disorders and Stroke | R01NS110945 | Matthew C Weston |
| National Institute of General Medical Sciences | P30 103498 | Matthew C Weston |

The funders had no role in study design, data collection and interpretation, or the decision to submit the work for publication.

### Author contributions

Matthew P McCabe, Kathryn A Laprade, Formal analysis, Investigation; Erin R Cullen, Software, Formal analysis, Validation, Investigation, Visualization, Methodology; Caitlynn M Barrows, Formal analysis, Investigation, Visualization; Amy N Shore, Investigation, Writing-Original Draft Preparation, Writing-Review and Editing; Katherine I Tooke, Investigation; James M Stafford, Formal analysis, Investigation, Writing - review and editing; Matthew C Weston, Conceptualization, Data curation, Formal analysis, Supervision, Funding acquisition, Validation, Investigation, Visualization, Methodology, Writing-Original Draft Preparation, Writing-Review and Editing, Project administration

### Author ORCIDs

Caitlynn M Barrows http://orcid.org/0000-0003-4696-9354
Matthew C Weston https://orcid.org/0000-0001-5558-7070

### Ethics

Animal experimentation: This study was performed in strict accordance with the recommendations in the Guide for the Care and Use of Laboratory Animals of the National Institutes of Health. All of the animals were handled according to approved institutional animal care and use committee (IACUC) protocols of the University of Vermont. The protocol was approved by the University of

Vermont's Research Protections Office (Protocol Number: 16-001). All animals were killed under iso-fluorane anesthesia, and every effort was made to minimize suffering.

## Decision letter and Author response
Decision letter https://doi.org/10.7554/eLife.51440.sa1
Author response https://doi.org/10.7554/eLife.51440.sa2

## Additional files

### Supplementary files
• Transparent reporting form

### Data availability
All data generated or analysed during this study are included in the manuscript and supporting files.

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
