## [Decision Letter]

**Acceptance summary:**

This paper examines the role of mTORC1 and mTORC2 in presynaptic function, specifically evoked synchronous and asynchronous release as well as spontaneous release. The authors show that inactivation of mTORC2 inhibits synaptic vesicle fusion while the inactivation of mTORC1 impairs synchronous release but enhances spontaneous and asynchronous vesicle release. This work provides novel insight into mechanisms that regulate synaptic vesicle release and builds on the growing literature demonstrating key functional differences between evoked and spontaneous release.

**Decision letter after peer review:**

Thank you for submitting your article "mTORC1 and mTORC2 regulate distinct aspects of glutamatergic synaptic transmission" for consideration by *eLife*. Your article has been reviewed by three peer reviewers, and the evaluation has been overseen by a Reviewing Editor and Olga Boudker as the Senior Editor. The reviewers have opted to remain anonymous.

The reviewers have discussed the reviews with one another and the Reviewing Editor has drafted this decision to help you prepare a revised submission.

Overall, the reviewers agreed that your study was performed to a high standard and clearly documented diverse but specific functions of mTORC1 and mTORC2 complexes in synaptic transmission. That said, they also identified critical weaknesses that need to be addressed in a revision.

Summary:

mTOR kinase signaling is important for neuronal growth regulation and neuronal network integrity, as demonstrated by neurological disorders associated with disruption in proper mTOR signaling. Previous work indicated that mTOR interference regulates various aspects of neuronal function and morphology, and may cause the neurological dysfunction through impaired synaptic functions. The mTOR kinase exists in two complexes, and the main focus of this manuscript is to examine a putative differential impact of mTORC1 and -2 complexes in synaptic transmission of central mammalian neurons. The authors use clever genetic deletion mTOR subcomplex components Raptor and Rictor. While previous studies showed that either complex is required for proper formation of dendritic structure, here the detailed functional and morphological analysis allowed the authors to unambiguously demonstrate that the two complexes perform partially overlapping and also quite distinct functions at the pre- and post synapse. This finding that Rictor/mTOR complexes predominantly act on presynaptic processes, while Raptor/mTOR predominately postsynaptic was quite surprising and has impact in understanding the differential roles of the mTOR kinase complexes in regulating synaptic strength in neuronal networks. The results are extensive in its depth of synaptic mechanisms (evoked, spontaneous release, synchronous vs asynchronous etc) and the outcome is quite clear, and proper statistical procedures were used. The controls are also adequate, and fully supporting the authors conclusion.

Essential revisions:

1) Given the divergent roles of mTORC1 and mTORC2 in regulating synaptic transmission, it is important to know whether there is any specificity to the subcellular localization of Raptor and Rictor. Is Raptor enriched postsynaptically and Rictor enriched presynaptically?

2) As discussed by the authors, a role for mTORC1 in regulating postsynaptic function is not particularly novel. Thus, the novel findings in the manuscript are that mTORC2 appears to be acting primarily to regulate presynaptic function. This finding is especially important because negative regulators upstream of mTOR are implicated in disorders of the brain. However, there are no experiments designed to determine how mTORC2 is regulating presynaptic function. We think it is important for the authors to show in at least one of their presynaptic measures a potential mechanism that can explain the effects of Rictor reduction.

3) Figure 1. Methods for verifying KO are rather indirect. Please provide a direct validation of the KO.

---

## [Author Response]

Essential revisions:1) Given the divergent roles of mTORC1 and mTORC2 in regulating synaptic transmission, it is important to know whether there is any specificity to the subcellular localization of Raptor and Rictor. Is Raptor enriched postsynaptically and Rictor enriched presynaptically?

Our failure to observe a specific immunofluorescence signal with available antibodies complicated our approach to this question. So, instead of directly localizing Raptor or Rictor at the synapse, we relied on levels of their downstream targets. Indeed, a previous publication (Tang et al, 2002) had shown that a well-known mTORC1 target, 4EBP, is localized at the postsynapse. We therefore performed new immunostaining experiments to show that the mTORC2 target pAkt is localized at the presynapse, and that this signal is significantly reduced in Rictor-KO neurons. The results of these experiments are now presented in a new Figure 6.

2) As discussed by the authors, a role for mTORC1 in regulating postsynaptic function is not particularly novel. Thus, the novel findings in the manuscript are that mTORC2 appears to be acting primarily to regulate presynaptic function. This finding is especially important because negative regulators upstream of mTOR are implicated in disorders of the brain. However, there are no experiments designed to determine how mTORC2 is regulating presynaptic function. We think it is important for the authors to show in at least one of their presynaptic measures a potential mechanism that can explain the effects of Rictor reduction.

We agree with the reviewers that additional insight into the mechanisms underlying mTORC2 regulation of presynaptic function would improve the manuscript. Because one of the minor points raised was the evidence for a change in the Ca^2+^ dependency of release, we decided to address this by testing whether mTORC2 inactivation alters the eEPSC-Ca^2+^ dose-response curve. We found that the EC50 for Ca^2+^ was significantly greater in Rictor-KO neurons, suggesting that the mechanism underlying the effect of mTORC2 inactivation on SV release is due to altered Ca^2+^ sensitivity. The results of these experiments are now presented in a new Figure 6.

3) Figure 1. Methods for verifying KO are rather indirect. Please provide a direct validation of the KO.

In our original submission, we relied on three indirect results as evidence that *Raptor* and *Rictor* were conditionally deleted in our neurons: expected reductions in the levels of known targets, expected reductions in soma size, and in dendritic outgrowth. The reasons we relied on this indirect approach were that available antibodies for Raptor and Rictor proteins gave a very poor, non-specific immunofluorescent signal, and we expected that the presence of WT astrocytes that express normal levels of Raptor and Rictor in our preparation would confound a bulk approach (e.g. PCR or Western blot analysis of lysates). We believe that these three measurements together provide strong evidence that we are efficiently deleting *Raptor* and *Rictor* from our cells, however, we recognize the value in providing a more direct measurement of their loss. In response to the comments we again attempted to optimize available antibodies for immunofluorescence, but failed to observe a specific signal. As an alternative approach, we plated a higher density of neurons on WT astrocytes and performed Western blot analysis. The experiments showed significant reductions in both Raptor and Rictor protein levels. Given the ratio of astrocytes to neurons in the culture and the efficiency of viral transduction (approx. 80%), this indicates a very high efficiency of recombination in transduced neurons. The results of this experiment are shown in Figure 1—figure supplement 1.